# ENACT: Evaluating Embodied Cognition with World Modeling of Egocentric Interaction

## Abstract

Embodied cognition argues that intelligence arises from continuous sensorimotor interaction with the world. Modern Vision-Language Models (VLMs), however, are trained disembodiedly on vast static datasets. We therefore ask: to what extent does embodied cognition emerge from such training? To investigate this, we introduce ENACT, a benchmark that probes this question through world modeling from egocentric interaction. Grounded in a partially observable Markov decision process (POMDP) framework, ENACT comprises two complementary sequence reordering tasks: forward world modeling (predicting an ordered sequence of future states from actions) and inverse world modeling (inferring an ordered sequence of actions from state changes). Correctly solving these tasks indicates that the model has a solid understanding of how the environment will evolve given one's actions. Our scalable dataset contains 8,972 QA pairs derived from diverse, long-horizon household activities in the BEHAVIOR simulator. Experiments reveal a significant performance gap between state-of-the-art VLMs and humans, which widens dramatically as interaction horizons lengthen. We find that models consistently solve the inverse problem better than the forward one and exhibit strong embodied biases, showing a preference for right-handed actions and performance degradation with camera perspectives that deviate from those of human vision. Code and supplementary materials are available in our anonymous repository.

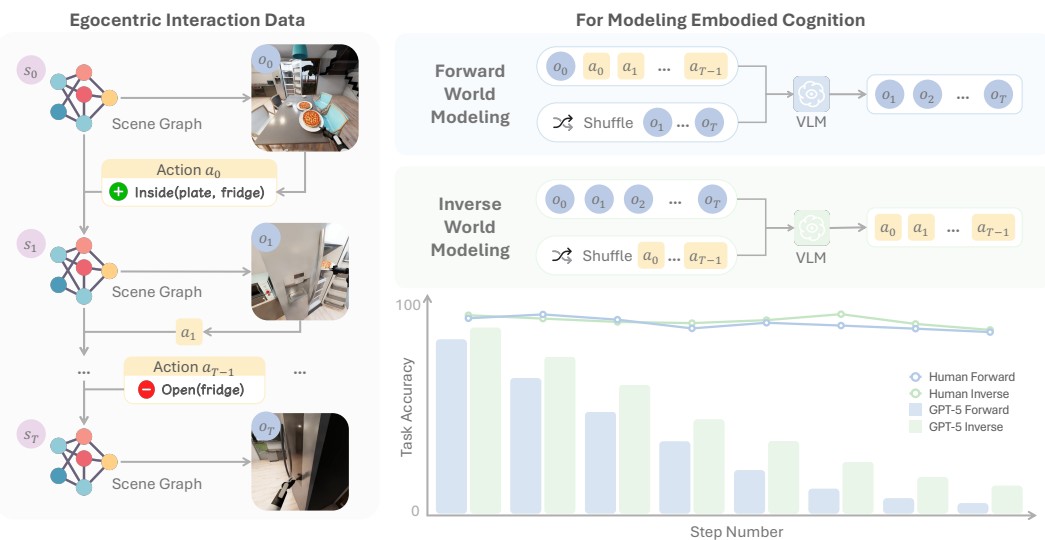

Figure 1: Grounded in a POMDP framework, **ENACT** probes embodied cognition in a **simple and scalable** way via world modeling through egocentric interaction (left). It poses two tasks (right top): **forward world modeling** (ordering observations given actions) and **inverse world modeling** (ordering actions given observations). Evaluation (right bottom) shows that GPT-5 performance drops as step length scales, solves better on inverse task, and lags behind humans.

## 1 Introduction

Intelligent behavior in the physical world involves grounding abstract knowledge in interaction with the environment. Embodied cognition argues that intelligence is not passively acquired, but *enacted*

through a continuous sensorimotor interaction with the world (Smith & Gasser, 2005). This process often integrates three key components: spatial perception of geometric structure, physical interaction for causal learning, and linguistic abstraction for generalization, as emphasized within embodied perspectives on cognition (Frick & Möhring, 2016; Thompson, 2005; Clark, 2006; Barsalou, 2020).

Despite the rapid progress in large foundation models like Vision-Language Models (VLMs) (OpenAI, 2025; DeepMind, 2025), these models are predominantly trained in a disembodied manner on internet-scale non-interactive data, with many more tokens than a human's lifetime experience. This raises a natural question: **to what extent does embodied cognition emerge from such training?** Existing benchmarks provide valuable insights by largely isolating one dimension at a time: spatial perception in static scenes (Ramakrishnan et al., 2024), physical interaction in contrived settings (e.g., a ball hits another ball in a clean environment; Yi et al., 2019; Gao et al., 2025), or linguistic reasoning and planning (Li et al., 2024b). However, a critical gap remains: studying these components in synergy within an embodied setting.

To this end, we introduce ENACT, a benchmark that probes embodied cognition through **world modeling via egocentric interaction**. A world model (Ha & Schmidhuber, 2018) captures an environment's dynamics. In the context of embodied agents, we focus on two forms: forward dynamics, which predict the next state from a given state and action, and inverse dynamics, which infer the action from a state transition. Grounded in a partially observable Markov decision process (POMDP, Åström, 1965, shown in Figure 1), we frame our tasks as *sequence reordering* to benchmark holistic understanding of world models and avoid designer biases. Specifically: In **forward world modeling**, given an initial observation and a sequence of abstract actions (symbolic scene graph (Johnson et al., 2017) state transitions), the model must reorder a shuffled sequence of future observations. In **inverse world modeling**, given an ordered sequence of observations, the model must reorder the corresponding shuffled action sequence. Built with the BEHAVIOR simulator (Li et al., 2024a), our dataset is scalable and well-suited for controlled experimentation, enabling probing of agent properties like long-term memory, the perception-action loop, embodied awareness, and data biases.

To ensure the data scalability of ENACT, we design a data generation pipeline. Starting with human demonstrations from BEHAVIOR, we replay them to generate trajectories with aligned symbolic scene graphs (states) and egocentric RGB observations. We represent actions with abstract state changes (e.g., `Remove Open(fridge)`). We then sample subsequences of desired lengths to assemble the key-frame trajectories for our QAs. Our dataset is scalable because our combinatorial sampling generates vast QAs from a single demonstration, and BEHAVIOR provides thousands of such demonstrations. We then use templates to build QAs for two world modeling tasks. We report two metrics at different granularities: Task Accuracy (exact ordering) and Pairwise Accuracy (fraction of adjacent pairs correctly ordered).

Our experiments reveal that ENACT is challenging for current VLMs, which lag significantly behind human performance (Figure 1). This performance gap widens as the task horizon increases, where VLM accuracy degrades sharply with trajectory length while human performance remains high. We also find that all models consistently perform better on inverse than forward world modeling. Furthermore, we uncover two notable biases: VLMs show a clear preference for understanding right-handed dynamics, and the performance of representative models like GPT-5 mini drops significantly with non-human-eye-like camera configurations.

Overall, our contributions are threefold: (1) We introduce ENACT, a benchmark with a scalable dataset for probing embodied cognition via forward and inverse world modeling from egocentric interaction. (2) We provide a large dataset of 8,972 QAs and a reproducible data generation pipeline using the BEHAVIOR simulator to create diverse, long-horizon embodied tasks at scale. (3) Through extensive experiments on state-of-the-art VLMs, we uncover their limitations in long-horizon embodied world modeling and reveal sensitivities to embodied data biases.

## 2 ENACT: EGOCENTRIC INTERACTIVE EMBODIED COGNITION TEST

### 2.1 PROBLEM FORMULATION

We investigate the embodied cognition of VLMs by framing it as a world modeling problem, which we probe using egocentric, interactive reasoning tasks. We formulate our benchmark from robot demonstration data, comprised of state-observation pairs $\{(s_t, o_t)\}$. The state $s_t$ is a symbolic scene

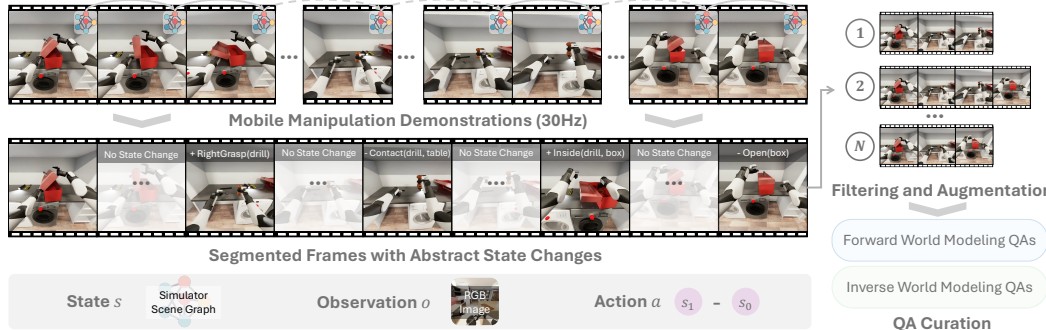

Figure 2: **Overview of ENACT data curation pipeline.** We first replay robot trajectories to obtain aligned scene graphs (states) and RGB observations. The raw trajectory is then segmented by identifying frames where an abstract state change occurs (i.e., the scene graph difference is non-empty). From this set of segmented frames, we sample multiple key-frame trajectories, which are finally used to construct the forward and inverse world modeling questions.

graph from the simulator state space $\mathcal{G}$, while the observation $o_t \in \mathbb{R}^{H \times W \times 3}$ is the corresponding egocentric RGB image. A symbolic scene graph is a structured data model that represents the objects in a scene as symbolic nodes (e.g., `On(fridge)`) and their relationships as edges (e.g., `OnTop(pen, desk)`). We view the underlying embodied task as a POMDP. As shown in Figure 2, we first filter this raw data to identify all timestamps where an abstract scene graph state change occurs (i.e., the scene-graph difference $\delta(s_t, s_{t-1}) \neq \varnothing$). This process yields a smaller, chronologically ordered set of segmented frames, which serve as the candidate pool for our benchmark.

From the pool of segmented frames, we sample $R$ trajectories, each with a chronologically ordered tuple $\pi = (i_0, \cdots, i_{L-1})$ of $L$ key frames. This initial abstraction into discrete decision epochs is similar to a semi-MDP (Sutton et al., 1999). However, we treat each of these final key-frame trajectories as a self-contained POMDP instance with scene graphs $S_\pi$ and observations $O_\pi$. For $k = 0, \cdots, L-2$, the action connecting consecutive key frames is the visible scene-graph delta $a_k := \Delta_{\text{Vis}}(s_{i_{k+1}}, s_{i_k})$, where $\Delta_{\text{Vis}}$ returns the subset of differences in $\delta(s_{i_{k+1}}, s_{i_k})$ that are visible in both images. Together, these actions form a discrete symbolic action space $\mathcal{A}$. For notation simplicity, we relabel indices in $\pi$ for each key-frame trajectory to $\pi = (0, \cdots, L-1)$ and $(s_k, o_k) := (s_{i_k}, o_{i_k})$.

Building on these trajectories, we formalize two tasks. For **forward world modeling**, given the current image $o_0$, the correct ordered action sequence $(a_0, \ldots, a_{L-2})$, and a *shuffled list* of next-state images $O' = (o'_1, \ldots, o'_{L-1})$, the model outputs a permutation $\sigma \in \text{Sym}([L-1])$ that orders the images to match the actions: $(o'_{\sigma(1)}, \ldots, o'_{\sigma(L-1)}) = (o_1, \ldots, o_{L-1})$. For **inverse world modeling**, given $o_0$, the correctly ordered state images $(o_1, \ldots, o_{L-1})$, and a *shuffled list* of actions $A' = (a'_0, \ldots, a'_{L-2})$, the model outputs a permutation $\tau \in \text{Sym}([L-1])$ that orders the actions to be consistent with the state progression: $(a'_{\tau(1)}, \ldots, a'_{\tau(L-1)}) = (a_0, \ldots, a_{L-2})$.

## 2.2 KEY-FRAME TRAJECTORIES SYNTHESIS FOR SCALABLE DATA GENERATION

**Segmented Frames with Abstract State Changes.** Raw robot trajectories often contain long stretches with no semantic changes (e.g., gripper motion when opening the toolbox in Figure 2). We mark a timestamp $t$ whenever the simulator state makes a minimal abstract state change (e.g., the robot is now grasping the drill with the right hand). The BEHAVIOR simulator exposes boolean and relational predicates, where flipping one predicate or updating a relation is our atomic state change. A time $t$ enters the candidate pool if the scene-graph difference $\delta(s_t, s_{t-1})$ is nonempty. To avoid near-duplicate frames, we compare each new change with the last accepted segmented frame: we form a predicate-level *change signature* $c_j$ and keep $t$ only if its cosine similarity with the previous signature $c_{j-1}$ is below a threshold. This yields a chronological set of segmented frames $\mathcal{K} = \{t_1 < \cdots < t_M\}$ with $(s_{t_i}, o_{t_i})$. Thresholds and further details are in the Appendix A.2.1.

**Key-Frame Trajectories Synthesis.** From the segmented $M$ frames, we sample length-$L$ key-frame trajectories $\pi = (i_0, \ldots, i_{L-1})$ with $1 \le i_0 < \cdots < i_{L-1} \le M$, so indices do not need to be adjacent. Each candidate is strictly validated: for every $k$, the visible state change $\Delta_{\text{Vis}}(s_{i_{k+1}}, s_{i_k})$ is nonempty,

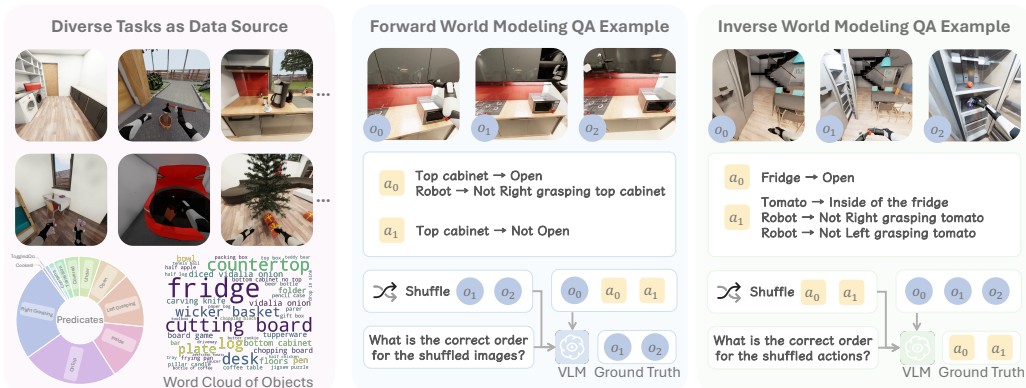

Figure 3: **Data sources and QA examples.** ENACT is built from diverse, long-horizon activities performed by real robots (left). We provide examples for (mid) forward world modeling and (right) inverse world modeling. More QA examples and prompts are available in the Appendix A.3.3.

and the edited objects are visible in both images, except for object transitioning events (e.g., pineapple being diced), where transient occlusion is permitted. We then treat each valid key-frame trajectory as an individual POMDP instance, with $S_\pi$ and $A_\pi$ as defined in the problem formulation. To make data generation *scalable*, we exploit that typically $L < M$ (in practice $L \leq 10$ while $M \gtrsim 30$), and we use skipping to convert trajectory construction into a *"seat selection"* combinatorics problem, choosing $L$ seats out of $M$, which yields at most $\binom{M}{L}$ distinct candidates from a single trajectory. The detailed algorithm is in the Appendix A.2.2.

**Reordering Questions Generation.** While multiple-choice questions suit single-step tasks, designing unbiased distractors for long-horizon interactions is infeasible. We therefore adopt *sequence reordering*, a simple formulation that avoids designer bias and requires a holistic understanding of the entire interaction. These trajectories are then converted into the forward and inverse world-modeling tasks by shuffling future states or actions, as specified in the problem formulation Section 2.1.

### 2.3 DATASET OVERVIEW AND EVALUATION DESIGN

**Dataset Overview.** We construct the benchmark from the BEHAVIOR simulator and challenge (Li et al., 2024a). BEHAVIOR Challenge provides diverse long-horizon household activities, and we select 29 activities from it and use one trajectory per activity to recover aligned pairs $\{(s_t, o_t)\}$. Each trajectory is segmented into *segmented frames* $\mathcal{K}$, then converted into key-frame trajectories and finally into two QA types: forward world modeling and inverse world modeling (examples in Figure 3). Across step lengths $L \in \{3, \dots, 10\}$ we sample about $560$ items per $L$ for each QA type, yielding 8,972 total questions. The data uses common predicate classes (e.g., `Open`, `Cooked`, `Grasping`), and distributions are shown in Figure 3. Full statistics appear in the Appendix A.3.1.

**Evaluation Design.** Multiple valid answers can exist for a given question. We therefore use an *online verifier* that accepts any predicted permutation, $\sigma$ or $\tau$, that is consistent with the corresponding input description constraints. Furthermore, we report two complementary metrics: *Task accuracy* captures exact ordering, while *Pairwise accuracy* grants partial credit for near-correct sequences. Specifically, (1) *Task accuracy* measures exact success at the question level. A question receives score 1 if the verifier accepts the full prediction and 0 otherwise. The dataset score is the average over questions, $\text{TA} = (1/|\mathcal{D}|) \sum_{x \in \mathcal{D}} \mathbf{1}\{\text{accepted}(x)\}$. (2) *Pairwise accuracy* measures stepwise consistency. For a question with length $L$, we count how many adjacent pairs pass the verifier's local check (state–action for forward; action–state for inverse) and divide by $L$. We report the micro-average across the split, $\text{PA} = \left(\sum_x \#\text{correct pairs in } x\right) / \left(\sum_x L_x\right)$, which is equivalent to averaging per-item pairwise scores when $L$ is fixed. Detailed evaluation implementation can be found in the Appendix A.3.2.

## 3 EXPERIMENTS AND ANALYSIS

### 3.1 WORLD MODELING AS A PROXY FOR EVALUATING EMBODIED COGNITION

**Experimental Setup.** (1) *VLM evaluation setup.* We evaluate ENACT with 7 proprietary VLMs from 3 families (OpenAI, 2025; DeepMind, 2025; Anthropic, 2025) and 22 open-weight models from 10 families (Wang et al., 2025; Bai et al., 2025; Hong et al., 2025; MetaAI, 2025; Team et al.,

Table 1: **Evaluation on ENACT (Pairwise Accuracy).** Dark gray indicates the best result within each category (Proprietary or Open-Weight Models), and Light gray denotes the second-best result within the category. Complete results are in the Table 7 (Task Accuracy) and 8 (Pairwise Accuracy).

| Model | Forward World Modeling | | | | | | | | Inverse World Modeling | | | | | | | |
|---|---|---|---|---|---|---|---|---|---|---|---|---|---|---|---|---|
| | 3 | 4 | 5 | 6 | 7 | 8 | 9 | 10 | 3 | 4 | 5 | 6 | 7 | 8 | 9 | 10 |
| *Proprietary Models* | | | | | | | | | | | | | | | | |
| GPT-5 | 84.62 | 75.26 | 69.96 | 64.18 | 57.48 | 52.16 | 49.45 | 46.93 | 86.28 | 80.37 | 76.09 | 68.78 | 65.71 | 62.13 | 57.12 | 55.33 |
| GPT-5 mini | 87.50 | 76.25 | 70.65 | 63.41 | 58.14 | 52.38 | 46.65 | 44.11 | 85.05 | 76.77 | 75.43 | 67.67 | 63.79 | 57.04 | 55.04 | 50.02 |
| GPT-5 nano | 67.83 | 50.29 | 38.61 | 30.35 | 25.97 | 21.90 | 17.59 | 16.84 | 72.81 | 53.95 | 42.48 | 36.45 | 31.68 | 28.20 | 24.11 | 20.33 |
| Gemini 2.5 Pro | 86.10 | 76.42 | 69.83 | 60.80 | 53.26 | 48.12 | 40.12 | 36.98 | 87.94 | 81.18 | 75.39 | 70.03 | 66.03 | 62.91 | 57.78 | 56.62 |
| Gemini 2.5 Flash | 81.64 | 67.94 | 54.17 | 43.38 | 37.43 | 32.73 | 29.88 | 28.07 | 82.78 | 72.18 | 60.83 | 58.19 | 53.14 | 51.78 | 47.99 | 44.98 |
| Gemini 2.5 Flash-Lite | 64.34 | 49.07 | 38.70 | 33.87 | 27.81 | 25.44 | 23.31 | 20.31 | 69.58 | 57.55 | 46.04 | 39.09 | 34.06 | 30.18 | 27.51 | 23.16 |
| Claude Sonnet 4 | 65.65 | 45.82 | 36.65 | 30.52 | 26.61 | 22.78 | 21.49 | 20.16 | 73.25 | 56.85 | 48.87 | 43.07 | 37.00 | 32.71 | 30.50 | 28.49 |
| *Open-Weight Models* | | | | | | | | | | | | | | | | |
| GLM-4.5V | 74.30 | 59.99 | 47.65 | 38.78 | 30.83 | 25.69 | 21.60 | 19.67 | 80.59 | 69.28 | 57.04 | 51.53 | 46.95 | 41.68 | 37.36 | 37.93 |
| Llama-4-Mav-17B-128E-Ins | 72.47 | 52.09 | 43.87 | 35.30 | 29.90 | 25.89 | 22.79 | 20.49 | 72.55 | 62.60 | 50.52 | 43.10 | 35.17 | 31.68 | 28.10 | 25.80 |
| InternVL3.5-241B-A28B | 75.79 | 62.25 | 50.83 | 45.85 | 37.84 | 32.88 | 27.85 | 25.24 | 82.26 | 70.09 | 60.61 | 53.38 | 45.90 | 39.35 | 34.12 | 30.56 |
| Gemma-3-27b-it | 63.29 | 44.66 | 32.04 | 25.82 | 22.11 | 19.50 | 16.74 | 16.29 | 64.95 | 48.37 | 40.04 | 33.87 | 28.53 | 23.63 | 21.74 | 19.36 |
| QVQ-72B-Preview | 69.14 | 52.96 | 40.83 | 36.27 | 33.16 | 30.63 | 26.30 | 24.76 | 71.33 | 58.77 | 48.43 | 44.36 | 40.26 | 39.30 | 36.66 | 36.58 |
| Qwen2.5-VL-72B-Ins | 78.15 | 60.05 | 49.87 | 41.92 | 36.77 | 31.73 | 28.03 | 25.07 | 77.80 | 65.85 | 53.30 | 48.19 | 44.07 | 37.57 | 33.76 | 36.27 |
| Qwen2.5-VL-32B-Ins | 67.83 | 55.46 | 44.35 | 35.75 | 27.52 | 26.42 | 22.01 | 18.07 | 63.55 | 59.70 | 54.57 | 51.01 | 49.36 | 47.17 | 41.47 | 40.16 |
| Ovis2.5-9B | 58.39 | 42.51 | 34.96 | 31.08 | 24.61 | 20.78 | 18.11 | 16.96 | 64.86 | 51.74 | 41.65 | 35.47 | 30.95 | 26.64 | 23.70 | 23.25 |
| MiniCPM-V-4.5 | 60.75 | 38.73 | 33.65 | 25.47 | 24.81 | 21.40 | 21.56 | 18.33 | 69.23 | 53.08 | 47.35 | 39.55 | 34.87 | 30.63 | 27.05 | 25.71 |
| Idefics3-8B-Llama3 | 60.23 | 36.99 | 31.83 | 24.25 | 21.29 | 20.80 | 20.46 | 17.71 | 47.38 | 33.86 | 27.26 | 23.48 | 19.87 | 18.50 | 17.04 | 15.16 |
| Cosmos-Reason1 | 56.28 | 41.86 | 34.75 | 28.40 | 26.46 | 26.49 | 25.41 | 24.88 | 58.30 | 45.93 | 44.25 | 38.50 | 35.72 | 34.56 | 31.50 | 28.64 |
| **Human Performance** | **93.62** | **95.30** | **95.04** | **93.87** | **95.43** | **95.41** | **94.75** | **95.13** | **92.05** | **93.56** | **94.35** | **94.25** | **95.96** | **97.74** | **96.30** | **96.29** |

2025; Lu et al., 2025; Yao et al., 2024; Azzolini et al., 2025; Team, 2024). For input, all images are resized to $512 \times 512$, and we use a unified prompt template per QA type. Models are instructed to return a parsable Python list encoding a permutation of indices. We apply the online verifier in Section 2.3 and report Task Accuracy and Pairwise Accuracy. (2) *Human evaluation setup.* We also recruit trained annotators to answer the benchmark under the same and instructions as the models. For inter-annotator agreement (IAA), we uniformly stratify 240 items over QA type and step length, and collect independent labels from three annotators. We report Krippendorff's $\alpha = 0.83$, indicating strong agreement among annotators. Full details are in the Appendix B.2.1 and B.1.1.

We visualize *Task Accuracy* for GPT-5 and human annotators in Figure 1. Since many models collapse at long horizons ($L = 8$–10, near-zero task success), we focus on the more informative *Pairwise Accuracy*. We show the main results in Table 1. Further analysis can be seen in Appendix B.2.2

**Is inverse world modeling easier than forward?** Across families and step lengths, inverse consistently outperforms forward, with the margin widening as $L$ grows. For example, GPT-5 and Gemini 2.5 Pro maintain clear gaps at $L \geq 6$, and open-weight models such as GLM-4.5V and Qwen2.5-VL also show higher inverse scores than forward for most $L$ (see Table 1).

**How does performance change with step length?** Accuracy decreases monotonically with $L$ for nearly every model, no matter proprietary or open-weight. Shorter tasks ($L \leq 4$) are manageable for several VLMs, while longer tasks ($L \geq 8$) are challenging even for the strongest models. Pairwise Accuracy slows down the performance drop compared to Task Accuracy, but follows the same trend.

**Can VLMs achieve near-human performance?** Human performance is far better than any evaluated VLM. SOTA VLMs like GPT-5 and Gemini-2.5 Pro achieve comparable performance with humans at step length 3, but the performance drops significantly when step length scales.

**What is the performance comparison among VLMs?** GPT-5 and Gemini 2.5 Pro are the strongest overall in both forward and inverse settings. Several open-weight VLMs are competitive: InternVL3.5-241B-A28B, GLM-4.5V, and Qwen2.5-VL often close much of the gap, and even surpass Claude 4 Sonnet in multiple settings (e.g., inverse at $L = 3$–6). Notably, GPT-5 mini is highly competitive, even achieving the best score in short and mid horizons (e.g., forward at $L = 3, 7, 8$).

> 💡 **Key Takeaways: World Modeling as a Proxy for Evaluating Embodied Cognition**
>
> - *Inverse consistently surpasses forward, and the margin grows as the horizon $L$ increases.*
> - *Accuracy declines steadily with step length $L$, and all VLMs drop sharper at long horizons.*

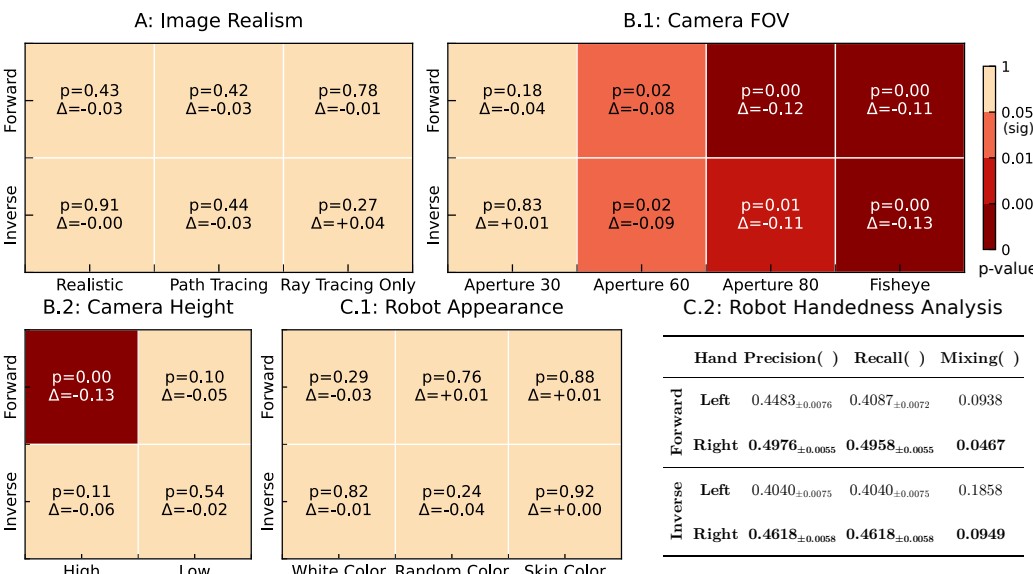

Figure 4: **Probing experiment results with GPT-5 mini on ENACT.** Heatmaps show two-tailed unpaired t-test results against the baseline, using *Pairwise Accuracy*. $p < 0.05$ is considered *significant*. Darker red means more significant. $\Delta$ is the performance change from the baseline. If *significant* and $\Delta < 0$, the setting is worse than the baseline. C.2 reports the robot's performance on the left- and right-hand predicates, where *Mixing* is the proportion of ground truth left or right cases that are predicted as the other hand (i.e., mixing one hand into the other). $\pm$ means standard error.

## 3.2 SENSITIVITY TO IMAGE REALISM

Since ENACT data is generated with the BEHAVIOR simulator. Despite being photo-realistic, we ask whether VLMs are sensitive to image realism, thus identifying if there is a sim-to-real gap.

**Experimental Setup.** (1) *Probing configuration.* We use GPT-5 mini as the base model for SOTA VLMs due to its strong cost-performance balance shown in Table 1. For diversity, we also evaluate InternVL3.5-241B and report its performance across all settings in Figure 17. We evaluate step lengths $L \in \{3, 6, 9\}$. For each $L$ and each QA type (forward, inverse), we sample 50 items, yielding 300 total QAs. We report, for each setting, the Pairwise Accuracy difference $\Delta = \mathrm{PA}_{\text{baseline}} - \mathrm{PA}_{\text{variant}}$ and two-tailed unpaired p-values versus the baseline. $|\Delta| < 0.05$ will be considered as a small change. (2) *Image realism implementation.* BEHAVIOR uses Isaac Sim (NVIDIA, 2025), our *baseline* uses Ray Tracing (NVIDIA, 2021) with default global effects. We probe three alternatives on a realism spectrum: *Realistic* (segmented frames translated to a real-world style using GPT-image-1 OpenAI (2025)), *Path Tracing* (higher-fidelity rendering, Kajiya (1986)), and *Ray Tracing Only* (Ray Tracing with global effects such as reflections and stage lights disabled). Detailed setup, prompts, and examples are in the Appendix B.4. Results are summarized in Figure 4 (panel A).

**Does rendering realism change performance?** We find no statistically significant degradation or improvement across the spectrum. All settings have $p \geq 0.2$ relative to the baseline, and observed deltas are small across both QA types and all step lengths (Figure 4, A; Figure 17, A). This suggests that both GPT-5 mini and InternVL3.5-241B are not sensitive to image realism in our embodied tasks.

> 💡 **Key Takeaways: Image Realism**
> • *GPT-5 mini and InternVL3.5-241B are robust to image realism variations on our tasks.*

## 3.3 SENSITIVITY TO CAMERA CONFIGURATIONS

VLMs are mostly trained on RGB images that mirror how humans typically see the world. However, different embodiments may have diverse camera configurations. We therefore test whether VLM performance is sensitive to camera configuration, i.e., if dataset bias is present.

**Experimental Setup.** (1) *Probing configuration.* We reuse the setup from Section 3.2. We use GPT-5 mini as the base VLM, and report InternVL3.5-241B in the Appendix B.5. (2) *Camera FOV.*

The baseline is Aperture 40. We probe Aperture 30, 60, 80, and Fisheye. Rendering and all other parameters are held fixed. (3) *Camera Height.* The baseline is $(1.75\,\text{m})$ high for eye-level view used in Behavior replays. We probe High $(+0.5\,\text{m})$ and Low $(-0.25\,\text{m})$. We choose $(-0.25\,\text{m})$ since a lower height will consistently make relevant objects invisible. Examples are in the Appendix B.5. Results are summarized in Figure 4 (panels B.1 and B.2).

**Does field of view matter?** Figure 4 (B.1) shows the results. A small change to Aperture 30 shows no significant difference from baseline ($p > 0.1$). Larger deviations substantially hurt performance. Aperture $60, 80$, and Fisheye are consistently and significantly worse than baseline across QA types and step lengths ($p \leq 0.01$). This suggests that the model performs better with human-like intrinsics.

**Does camera height matter?** As shown in Figure 4 (B.2), increasing the camera height (*High*) significantly reduces GPT-5 mini's accuracy in the forward setting with $\Delta = -0.13$. By contrast, the *High* inverse setting shows no statistically significant change, but with a notable performance drop $\Delta = -0.06$. For the *Low* camera, both forward and inverse are not significantly different from the baseline, which may be because the $-0.25$ m is still within the normal human height distribution.

> 💡 **Key Takeaways: Camera Configurations**
> - *Models perform best on images that resemble what humans typically see.*
> - *Large apertures, a fisheye lens, and a high camera will harm models' performance greatly.*

### 3.4 DO VLMS HAVE EMBODIED BIASES?

To further understand the nature of VLM embodiment, we investigate two potential biases: **self-awareness** regarding the robot's own body and **handedness asymmetry**, a common trait in humans.

**Experimental Setup.** We probe these two aspects using distinct experimental setups. (1) *Robot Appearance*. To test for self-awareness, we assess whether VLMs can recognize their embodiment regardless of its appearance. We reuse the probing configuration from Section 3.2, with GPT-5 mini as the base model. The baseline is the default black-and-white robot appearance. We test three variants: White Color, Random Color (robot color is randomized at each frame), and Skin Color (robot is rendered with a human-like skin tone). (2) *Handedness Asymmetry*. Inspired by human motor control, where approximately 89% of the population is right-handed (Papadatou-Pastou et al., 2020), we investigate if VLMs exhibit a similar "dominant hand". We analyze this configuration with a predicate-level error analysis of all tested VLMs and report GPT-5 mini in Figure 4. We isolate all errors related to the `LeftGrasping` and `RightGrasping` predicates. Using the framework described in Section 3.5, we frame our metrics in terms of *Precision* and *Recall*. We also report *Mixing Rate*, which measures the proportion of ground-truth state differences for one hand that the model incorrectly attributes to the other. Higher precision and recall with lower mixing indicate greater proficiency. Appearance examples and handedness analysis are in the Appendix B.6.1 and B.6.2.

**Are VLMs aware of their own embodiment, and is this awareness robust to changes in their visual appearance?** As shown in Figure 4 and Figure 17 (panel C.1), altering the robot's appearance has no statistically significant impact on performance for both GPT-5 mini and InternVL3.5-241B. For all variants (White, Random, Skin Color), the performance deltas are small ($|\Delta| < 0.05$) and the results are not significant (all $p > 0.10$). This suggests that the model's understanding of its interaction with the world is not tied to a specific visual representation of its own body.

**Do VLMs exhibit a handedness asymmetry in their interactions with the world?** Our analysis of hand-related errors, summarized in Figure 4 (panel C.2), reveals a consistent and strong asymmetry (complete error results are shown in Figure 38a and 38b). For both forward and inverse tasks, the right hand consistently outperforms the left hand across all metrics. Precision and recall are substantially higher for the right hand, while the mixing rate is significantly lower. For instance, in the forward task, 9.38% of true left-hand changes were incorrectly identified as right-hand changes, whereas only 4.67% of right-hand changes were misattributed to the left. Full analysis is in Appendix C.1.

> 💡 **Key Takeaways: Embodied Biases**
> - *GPT-5 mini and InternVL3.5-241B are robust to the robot's appearance.*
> - *VLMs exhibit a significant right-handed bias, which is similar to human handedness.*

### 3.5 ERROR ANALYSIS

#### 3.5.1 PREPARATION FOR ERROR ANALYSIS

To gain a deeper insight into the reasoning failures of VLMs, we designed a systematic error analysis framework. Evaluating errors directly from output permutations (e.g., comparing predicted order [3, 2, 1] to ground truth [2, 3, 1]) is difficult and often uninformative about the underlying cognitive mistakes. Our approach instead converts the model's output into a format that allows for a direct, fine-grained comparison with the ground truth. For the **forward world modeling** task, we take the model's predicted permutation of images $(o'_{\sigma(1)}, \ldots, o'_{\sigma(L-1)}) = (o_1, \ldots, o_{L-1})$ and compute the corresponding sequence of actions (i.e., visible state differences) that this ordering implies: $\hat{a}_k := \Delta_{\text{Vis}}(s'_{\sigma(k+1)}, s'_{\sigma(k)})$. This yields a predicted action sequence $(\hat{a}_0, \cdots, \hat{a}_{L-2})$. For the **inverse world modeling** task, the model already outputs a predicted action sequence.

With both a predicted and a ground-truth action sequence, we can perform a pairwise comparison at each step $k$. Each action $a_k$ is a *set* of atomic state differences (e.g., {Add Open(fridge), Remove Inside(basket, cabinet)}). By comparing the predicted set $\hat{a}_k$ with the grounded-truth set $a_k$, we can categorize each atomic state difference. This comparison, similar to analyzing a Venn diagram, yields three primary outcomes for each ground-truth state difference: (1) *Correct*: The state difference is present in both the ground-truth and predicted sets. (2) *Omission*: The state difference is in the ground-truth set but missing from the prediction. (3) *Hallucination*: The state difference is in the predicted set but not in the ground truth. Detailed setup is in the Appendix C

We assume each state difference is an *independent event* and aggregate these counts across all actions and all questions in the dataset. Based on this framework, we classify errors into five main categories:

1. **Entity Substitution.** The model correctly identifies the state change predicate but applies it to the wrong object(s).

2. **Polarity Inversion.** The model correctly identifies both the object(s) and the predicate, but reverses the polarity of the change (e.g., 'remove' instead of 'add').

3. **Predicate Substitution.** The model correctly identifies the object(s) involved but describes the state change with an incorrect predicate.

4. **Hallucination.** The model predicts a state change that did not occur in the ground truth.

5. **Omission.** The model fails to predict a ground-truth state change that occurred.

#### 3.5.2 ERROR DISTRIBUTION ANALYSIS

Our error analysis for GPT-5, shown in Figure 5, reveals that the vast majority of errors fall into two main categories: **Omission** and **Hallucination**. For the forward task, these two error types account for a combined 81% of all failures. This figure is even higher for the inverse task, where they make up nearly 84% of errors. This indicates that the model's primary challenge is not misinterpreting the specifics of a known state change, but rather correctly identifying which changes occurred and which did not.

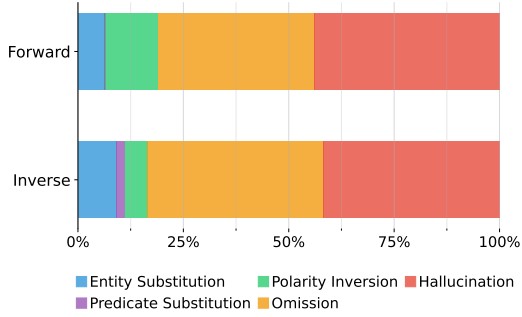

Figure 5: GPT-5 Error distribution across ENACT, broken down by forward and inverse tasks.

While Omission and Hallucination errors are dominant in both settings, their distribution shifts between tasks. In forward modeling, **Hallucination** is the most common error at 43.9%, followed by **Omission** at 37.1%. Remarkably, in the inverse task, these two errors are perfectly balanced, each accounting for exactly 41.8% of all failures. Other error types are far less frequent. **Polarity Inversion** is more common in the forward setting (12.4%) than the inverse (9.2%). Interestingly, **Entity Substitution** is also slightly more prevalent in the forward task (6.3% vs. 5.4%). Finally, **Predicate Substitution** remains the rarest error type, though it is more pronounced in the inverse setting (1.9%) compared to the forward task (0.3%). Detailed analysis can be found in the Appendix C.

## 4 RELATED WORK

**Embodied Cognition.** The theory of *Embodied Cognition* argues that intelligence arises from an agent's sensorimotor interaction within its environment, grounding abstract knowledge in perception and action (Gibson, 2014; Varela et al., 2017; Clark, 1998; Brooks, 1991; O'regan & Noë, 2001; Barsalou, 1999; Lakoff & Johnson, 2008). Within this framework, our work focuses on the integration of three key components: spatial perception, physical interaction, and linguistic abstraction (Frick & Möhring, 2016; Thompson, 2005; Clark, 2006; Barsalou, 2020). We propose to test these core components of embodied cognition through world modeling via egocentric interaction.

**World Modeling.** World models learn action-conditioned dynamics for imagination and planning (Ha & Schmidhuber, 2018; Hafner et al., 2019), achieving scalable gains from counterfactual rollouts (Hafner et al., 2023; Bruce et al., 2024; Agarwal et al., 2025; Janner et al., 2022). However, their grasp of embodied interaction remains limited, often due to a lack of physical grounding from training on internet data (Bruce et al., 2024; Agarwal et al., 2025) or a failure to maintain causal state progression (Finn & Levine, 2017; Ebert et al., 2018). Existing benchmarks reflect this gap, often scoring superficial qualities (Tian et al., 2023; Chi et al., 2024; Yue et al., 2025), remaining non-interactive (Bakhtin et al., 2019; Yi et al., 2019; Bear et al., 2021; Tung et al., 2023; Li et al., 2024a; Yang et al., 2025), or overlooking the consequences of individual actions (Qin et al., 2024; Chen et al., 2025). As argued by Xing et al. (2025), a world model should serve as a sandbox for reasoning. Our benchmark is therefore designed to probe forward and inverse ordering with a clean action space and scalable construction.

**VLMs in Embodied AI.** VLMs are central to embodied agents, acting as high-level planners (Ahn et al., 2022; Huang et al., 2023b; 2022; Liang et al., 2022a; Huang et al., 2023a; 2024) or end-to-end policies (Zitkovich et al., 2023; Kim et al., 2024; Team et al., 2024; Driess et al., 2023). However, current applications are often confined to tabletop manipulation or simulated settings with limited real-world execution (Lynch et al., 2023). Correspondingly, benchmarks tend to prioritize simple instruction-following, neglecting the multi-step, consequence-aware reasoning essential for complex interaction (Das et al., 2018; Padmakumar et al., 2022; Mees et al., 2022; Fan et al., 2022; Li et al., 2024b; Yang et al., 2025). We address this gap by introducing a benchmark that uses egocentric interaction to specifically probe an agent's understanding of forward and inverse world modeling. We include additional related work discussion in the Appendix D.

## 5 CONCLUSIONS AND LIMITATIONS

**Conclusions.** In this work, we introduced ENACT, a benchmark designed to evaluate the extent to which embodied cognition emerges in VLMs trained on static datasets. By framing our evaluation through the lens of forward and inverse world modeling from egocentric interaction, ENACT probes a model's understanding of environmental dynamics and the consequences of its actions. Grounded in a POMDP, we evaluate two types of sequence reordering tasks: forward world modeling, which predicts an ordered sequence of future states from actions, and inverse world modeling, which infers an ordered sequence of actions from state changes. Our extensive experiments reveal a significant performance gap between state-of-the-art VLMs and humans, a gap that widens dramatically as the interaction horizon increases. We consistently found that models solve the inverse problem more effectively than the forward one. Furthermore, our analysis uncovered strong embodied biases within these models, including a preference for right-handed actions and a significant performance drop with non-human-like camera perspectives. An in-depth error analysis showed that reasoning failures are primarily driven by the omission and hallucination of state changes. ENACT provides a scalable and insightful tool for charting a course toward more genuinely embodied artificial intelligence.

**Limitations.** Our work has limitations primarily related to its scope. First, while we introduce several probing tasks that reveal key model biases, this set is not exhaustive. The experiments on factors like camera configuration and agent appearance serve as foundational examples, but the ENACT framework is designed to be an extensible tool. It can support future, more complex investigations into a much broader spectrum of different embodied-related settings. Second, due to the significant computational cost of our evaluation, the in-depth probing experiments were necessarily focused on a representative subset of models and data. A broader evaluation across more architectures and larger data scales would be beneficial to generalize our findings.

ETHICS STATEMENT

The ENACT benchmark was generated in the BEHAVIOR simulator to avoid the privacy risks associated with real-world human data; it contains no human subjects or personally identifiable information. All human annotators hired for evaluation were compensated at rates significantly exceeding their local minimum wage and were not exposed to any sensitive content.

We acknowledge that the simulator may not fully capture the complexity of real-world environments, which can introduce biases and limit the generalizability of our findings. Furthermore, the large-scale models we evaluate carry a significant computational and environmental cost. While ENACT is intended for academic research, we recognize that the technologies it helps develop could have dual-use applications.

REPRODUCIBILITY STATEMENT

To ensure full reproducibility, our complete codebase is available at Anonymous Repository. This repository contains all scripts for data generation using the BEHAVIOR simulator (Li et al., 2024a), evaluation of all 29 Vision-Language Models, and analysis. Our implementation includes the automated verifier, prompt templates, and the code to replicate our main experiments, controlled probing studies (Sections 3.2, 3.3, and 3.4), and human baseline evaluation. The full ENACT dataset will be publicly released upon publication.

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

# Appendix

## Table of Contents

# A ENACT: EGOCENTRIC INTERACTIVE EMBODIED COGNITION TEST

## A.1 NOTATIONS

We list all the notations we used across the entire paper in the following two tables.

Table 2: Notation used throughout the paper.

| Notation | Short description | Notation | Short description |
|---|---|---|---|
| $T$ | # frames in a raw replay | $H, W$ | image height and width |
| $[M]$ | index set $\{1, 2, \ldots, M\}$ | $M$ | # segmented key frames |
| $\mathcal{K}$ | segmented timestamps $\{t_1 < \cdots < t_M\}$ | $(o_i, s_i)$ | RGB & scene graph at timestamp $t_i$ |
| $o_t$ | RGB image at time $t$ | $s_t$ | scene graph at time $t$ |
| $\mathcal{G}$ | space of scene graphs | $L$ | target trajectory length (steps) |
| $R$ | # sampled trajectories | $E$ | adjacency matrix on $[M]$ (DAG) |
| $\mathrm{Adj}(i)$ | successors of node $i$ | $|E|$ | # edges in the DAG |
| $\delta(\cdot, \cdot)$ | scene-graph difference (see long) | $\mathrm{Vis}(\cdot)$ | visibility predicate (see long) |
| $\Delta_{\mathrm{vis}}(\cdot, \cdot)$ | visible-change extractor (see long) | $\mathcal{A}$ | action space |
| $a_i$ | local action $s_{i+1} - s_i$ | $a_{i \to j}$ | action from $i$ to $j$ |
| $\pi$ | key-frame trajectory (see long) | $S_\pi$ | state sequence along $\pi$ |
| $A_\pi$ | action sequence along $\pi$ | $\Pi$ | set of sampled trajectories |
| $DP[\ell, i]$ | # paths of length $\ell$ ending at $i$ | $w_i$ | end-node weight $DP[L, i]$ |
| $\mathcal{P}$ | predecessor set in backtracking | $\mathrm{Categorical}(w)$ | weighted discrete distribution |
| $\mathbf{1}\{\cdot\}$ | Iverson bracket (true=1, false=0) | $\mathcal{D}$ | datasets |
| $c$ | component in signature $a_i^{sig}$ | $\gamma, \text{transition}$ | The operation key in component |
| $e$ | The entity involved in component | $\rho$ | The predicate in component |

| Notation | Longer description |
|---|---|
| $\mathcal{T} = \{(o_t, s_t)\}_{t=1}^{T}$ | Raw replay trajectory with RGB observations $o_t \in \mathbb{R}^{H \times W \times 3}$ and scene graphs $s_t \in \mathcal{G}$. |
| $\delta(s_i, s_j)$ | A difference operator over scene graphs summarizing semantic changes (objects, relations, attributes) between frames $i$ and $j$. |
| $\mathrm{Vis}(\delta(s_i, s_j))$ | Predicate returning 1 iff the semantic difference is visually verifiable; induces an edge $i \to j$ when $i < j$ and the predicate is true (frame skipping allowed). |
| $\Delta_{\mathrm{vis}}(s_i, s_j) \in \mathcal{A} \cup \{\varnothing\}$ | Action-level representation extracted from $\delta(s_i, s_j)$; may be atomic or composite and can be empty when no visible semantic change exists. |
| $\pi = (i_1, \ldots, i_L)$ | Key-frame trajectory: strictly increasing indices with valid edges $E_{i_\ell, i_{\ell+1}} = 1$ for all $\ell = 1, \ldots, L-1$. |
| $S_\pi, A_\pi$ | Sequences induced by $\pi$: $S_\pi = (s_{i_1}, \ldots, s_{i_L})$, $A_\pi = (a_{i_1 \to i_2}, \ldots, a_{i_{L-1} \to i_L})$. |
| $DP[\ell, i]$ recurrence | Base: $DP[1, i] = 1$. Recurrence: $DP[\ell, i] = \sum_{j < i} DP[\ell-1, j] \cdot E_{ji}$ for $\ell = 2, \ldots, L$. |
| $\mathcal{M}_\pi = \langle \{s_{i_\ell}\}, \{a_{i_\ell \to i_{\ell+1}}\}, P \rangle$ | Deterministic finite-horizon fragment induced by $\pi$ with transition $P(s_{i_\ell}, a_{i_\ell \to i_{\ell+1}}) = s_{i_{\ell+1}}$. |
| $a_i^{sig}$ | Signature corresponding to an action $a_i$, transformed from natural language to predicate-based structural format. |

## A.2 KEY-FRAME TRAJECTORIES SYNTHESIS FOR SCALABLE DATA GENERATION

### A.2.1 SEGMENTED FRAMES WITH ABSTRACT STATE CHANGES

Our motivation for segmenting raw robot trajectories stems from their inherent semantic sparsity. For instance, in a task like opening a toolbox, hundreds of consecutive frames (at a 30Hz) may pass

without any meaningful semantic change. Since we define our fundamental actions as abstract state changes, our objective is to isolate precisely those frames where such changes occur. This process ensures that we focus on the semantically significant moments of the interaction.

The BEHAVIOR simulator provides an RGB image and a corresponding scene graph, composed of nodes and edges, for every frame. We define an atomic action as a difference in the scene graph between two consecutive moments in time. These differences are categorized into three types: the **addition** of a new node or relation, the **removal** of an existing one.

We provide examples of a scene graph and our scene graph differences for two adjacent segmented frames, shown in Figure 6 and 7.

Our frame selection process is iterative. For a previously selected key-frame at time $t_{i-1}$, we search for the earliest subsequent frame $t_k$ that satisfies a set of criteria designed to ensure semantic significance and visual clarity.

First, to handle discrepancies where the rule-based simulator updates the scene graph before a change is visually apparent (e.g., registering an object as 'OnTop' upon initial contact), we introduce a temporal stability filter. A state change is only considered a candidate if the resulting new state persists for a minimum of 40 frames. This ensures the visual representation has stabilized and reflects the symbolic change.

Second, to prevent the recording of minor, oscillatory state changes, such as those that might occur from vibrations when a robot carries an object (e.g., a plate with a pizza), we employ a filtering algorithm to suppress these small fluctuations in the scene graph.

Finally, to ensure that each selected key-frame represents a sufficiently distinct change from the previous one, we implement a similarity check. We convert the scene graph difference between the last selected frame $t_{i-1}$ and a candidate frame $t_k$ into a one-hot vector, which serves as a unique signature for that state change. We then compute the cosine similarity between the signature of the change at $t_k$ and the signature of the previously accepted change at $t_{i-1}$. We aim to find a balance between maximizing the number of segmented frames and ensuring each frame depicts a clearly visible state change. Through empirical evaluation, we determined a cosine similarity threshold of **0.97**. A candidate frame $t_k$ is accepted only if its change signature's similarity to the previous one is below this threshold. This method effectively filters out near-duplicate frames while retaining a rich, sequential set of key-frames that clearly chronicle the task's progression.

### A.2.2 KEY-FRAME TRAJECTORIES SYNTHESIS

Given the set of $M$ segmented frames from the previous stage, the goal of Key-Frame Trajectory Synthesis (KFTS, see Algorithm 1) is to efficiently sample a large number of valid trajectories of a fixed length $L$. A trajectory is defined as a sequence of indices $\pi = (i_1, \ldots, i_L)$ such that $1 \leq i_1 < i_2 < \cdots < i_L \leq M$. The key constraint is that for any two consecutive frames $i_k$ and $i_{k+1}$ in the trajectory, the state change between them must be semantically meaningful and visually verifiable. The KFTS algorithm, detailed in Algorithm 1, accomplishes this efficiently by converting the problem into path sampling on a Directed Acyclic Graph (DAG) and using dynamic programming. The process consists of three main stages:

1. **Directed Acyclic Graph (DAG) Construction:** We first model the relationships between all segmented frames. The $M$ frames are treated as nodes in a graph. A directed edge exists from frame $i$ to frame $j$ (where $i < j$) if and only if the state difference $\delta(s_i, s_j)$ constitutes a valid, visible transition. This validity is determined by a predicate $\text{Vis}(\cdot)$, which checks if the objects involved in the state change are clearly visible in both frames, as described in Section 2.2. This process results in an adjacency matrix $E$ for a DAG, where $E_{ij} = 1$ indicates a valid one-step transition from frame $i$ to $j$.

2. **Dynamic Programming Path Counting:** Instead of enumerating all possible $\binom{M}{L}$ combinations, we use dynamic programming (DP) to efficiently count the number of valid trajectories. We build a DP table where $DP[\ell, i]$ stores the total number of valid trajectories of length $\ell$ that terminate at frame $i$. The base case is $DP[1, i] = 1$ for all frames $i$, as any single frame is a

```
A Scene Graph Example

{
  'nodes': [
    {'name': 'robot_r1', 'category': 'agent', 'states': []},
    {'name': 'plate_94', 'category': 'plate', 'states': []},
    {'name': 'plate_93', 'category': 'plate', 'states': []},
    {'name': 'bowl_92', 'category': 'bowl', 'states': []},
    {'name': 'bowl_91', 'category': 'bowl', 'states': []},
    {'name': 'pizza_90', 'category': 'pizza', 'states': []},
    {'name': 'pizza_89', 'category': 'pizza', 'states': []},
    {'name': 'floors_zqjkvm_0', 'category': 'floors', 'states': []},
    {'name': 'breakfast_table_xftrki_0', 'category': 'breakfast_table', 'states': []},
    {'name': 'fridge_petcxr_0', 'category': 'fridge', 'states': ['Open']},
    {'name': 'drop_in_sink_lkklqs_0', 'category': 'drop_in_sink', 'states': []},
    {'name': 'straight_chair_uofiqj_0', 'category': 'straight_chair', 'states': []},
    {'name': 'bottom_cabinet_rhdbzv_0', 'category': 'bottom_cabinet', 'states': []}
  ],
  'Edges': [
    {'from': 'robot_r1', 'to': 'plate_93', 'states': ['RightGrasping']},
    {'from': 'plate_94', 'to': 'pizza_90', 'states': ['Under']},
    {'from': 'plate_94', 'to': 'breakfast_table_xftrki_0', 'states': ['OnTop']},
    {'from': 'bowl_92', 'to': 'breakfast_table_xftrki_0', 'states': ['OnTop']},
    {'from': 'bowl_91', 'to': 'breakfast_table_xftrki_0', 'states': ['OnTop']},
    {'from': 'pizza_90', 'to': 'plate_94', 'states': ['OnTop']},
    {'from': 'pizza_89', 'to': 'plate_93', 'states': ['OnTop']},
    {'from': 'breakfast_table_xftrki_0', 'to': 'plate_94', 'states': ['Under']},
    {'from': 'breakfast_table_xftrki_0', 'to': 'floors_zqjkvm_0', 'states': ['OnTop']},
    {'from': 'straight_chair_uofiqj_0', 'to': 'floors_zqjkvm_0', 'states': ['OnTop']}
  ]
}
```

Figure 6: A scene graph representation detailing the entities (**nodes**) and their semantic or physical connections (**edges**) within the BEHAVIOR (Li et al., 2024a) environment.

```
A Scene Graph Difference Example

'2442':
{
  'type': 'diff',
  'add': {
    'nodes': [],
    'edges': [
      {'from': 'robot_r1', 'to': 'plate_93', 'states': ['RightGrasping']}
    ]
  },
  'remove': {
    'nodes': [],
    'edges': [
      {'from': 'plate_93', 'to': 'pizza_89', 'states': ['Under']},
      {'from': 'plate_93', 'to': 'breakfast_table_xftrki_0', 'states': ['OnTop']}
    ]
  }
}
```

Figure 7: An example of a scene graph difference, representing a state change by specifying the **added** and **removed** edges between objects.

---

**Algorithm 1:** KFTS: Key-Frame Trajectory Sampling

---

**Input:** Segmented frames $\{(o_i, s_i)\}_{i=1}^M$, step length $L \geq 2$, samples $R$, predicate Vis
**Output:** Set of key-frame trajectories $\Pi$
**Build DAG**: **for** $1 \leq i < j \leq M$ **do**
  $\lfloor\ E_{ij} \leftarrow [\, \text{Vis}(\delta(s_i, s_j))\,]$
**DP counting**: initialize $DP[1, i] \leftarrow 1$; **for** $\ell = 2..L$ **do**
  $\quad$ **for** $i = 1..M$ **do**
  $\quad\ \lfloor\ DP[\ell, i] \leftarrow \sum_{j<i} DP[\ell - 1, j] \cdot E_{ji}$

**Weights**: $w_i \leftarrow DP[L, i]$; **if** $\sum_i w_i = 0$ **then**
  $\lfloor$ **return** $\emptyset$

**Weighted backtracking sampling**: $\Pi \leftarrow \emptyset$; sample $R$ end-nodes $i_L^{(r)} \sim \text{Categorical}(w)$
**for** $r = 1..R$ **do**
  $\quad \pi \leftarrow [\, i_L^{(r)}\,]$, $cur \leftarrow i_L^{(r)}$
  $\quad$ **for** $\ell = L..2$ **do**
  $\quad\quad \mathcal{P} \leftarrow \{\, j < cur \mid E_{j,cur} = 1 \wedge DP[\ell - 1, j] > 0\,\}$
  $\quad\quad$ **if** $\mathcal{P} = \emptyset$ **then**
  $\quad\quad\ \lfloor$ **break**
  $\quad\quad$ sample $j^\star \in \mathcal{P}$ with prob $\propto DP[\ell - 1, j]$; prepend $j^\star$ to $\pi$; $cur \leftarrow j^\star$
  $\quad$ **if** $|\pi| = L$ **then**
  $\quad\ \lfloor$ add $\pi$ to $\Pi$
**return** $\Pi$

---

valid path of length one. The table is filled using the recurrence:

$$DP[\ell, i] = \sum_{j<i} DP[\ell - 1, j] \cdot E_{ji}$$

This equation sums the number of valid paths of length $\ell - 1$ ending at any valid predecessor $j$ of frame $i$. After filling the table up to length $L$, the entry $DP[L, i]$ gives the exact number of distinct, valid, length-$L$ trajectories that end at frame $i$.

3. **Weighted Backtracking Sampling:** With the DP table computed, we can sample trajectories efficiently without bias. To generate one trajectory, we first sample an end-node $i_L$ from all possible frames $\{1, \ldots, M\}$. The sampling is weighted, with the probability of selecting frame $i$ being proportional to its weight $w_i = DP[L, i]$. This ensures that frames that can be part of more trajectories are more likely to be chosen as endpoints.

Once the end-node $i_L$ is selected, we reconstruct the path backwards. To select the previous node $i_{L-1}$, we consider all valid predecessors $j$ of $i_L$ (i.e., all $j < i_L$ where $E_{j,i_L} = 1$). We sample the predecessor $j^\star$ with a probability proportional to $DP[L - 1, j^\star]$. This process is repeated iteratively: to find node $i_k$, we sample from the predecessors of $i_{k+1}$ with probabilities proportional to the values in the $DP[k, \cdot]$ row. This weighted backtracking ensures that every valid trajectory of length $L$ has a chance of being sampled, and the likelihood of sampling any specific path is uniform across all valid paths. We repeat this procedure $R$ times to generate the desired number of trajectories.

This DP-based approach is highly scalable as its complexity is polynomial in $M$ and $L$, making it far more efficient than a brute-force combinatorial search, especially when $M$ is large.

**Justification for Symbolic Scene Graphs.** We opt for a symbolic scene graph representation, and while it may not capture the fine-grained details of low-level motions, this abstraction is advantageous for our objectives for two primary reasons. First, our focus is on detecting semantic changes within a scene, a task that naturally aligns with the conceptual understanding capabilities of VLMs. VLMs excel at reasoning about objects, their states, and relationships, rather than continuous motor trajectories. Second, the symbolic predicates we employ are not merely abstract; they are grounded in practical robotics applications. This is demonstrated through our experiments with a real-world robot (Robot R1 Pro), confirming the real-world relevance of our chosen states. The feasibility of this symbolic approach is further substantiated by its use in guiding the data collection for the

BEHAVIOR benchmark, where these predicates defined the goal conditions for simulated activities and enabled human annotators to clearly verify whether task states were successfully achieved.

### A.2.3 QA GENERATION

The core of our benchmark is to evaluate embodied cognition through tasks that require an understanding of world dynamics. Traditionally, world models are evaluated on their ability to predict **forward dynamics** (i.e., given a state and an action, predict the next state) or infer **inverse dynamics** (i.e., given an initial and resulting state, infer the action that caused the transition). These evaluations are often instantiated as simple one-step, multiple-choice questions.

Our approach deliberately departs from this paradigm by framing the tasks as multi-step, sequential ordering problems. Instead of predicting a single outcome, the model must correctly sequence a whole trajectory of observations or actions. This design provides a more comprehensive test of an agent's reasoning capabilities. As such, correctly sorting a sequence of images or actions indicates that the model not only understands the basic inverse and forward dynamics of the world, but can also leverage that understanding to reason about action consequences in long-horizon settings. This ability to comprehend causal chains is a critical component of embodied cognition.

The generation of each question-answering (QA) pair follows a structured pipeline, beginning with a sampled key-frame trajectory $\pi$. The process is as follows:

1. **Action Sequence Generation.** For the given key-frame trajectory $\pi$, we first compute the corresponding action sequence, $A_\pi$. Each action in this sequence represents the symbolic scene graph difference between two consecutive key-frames. These symbolic differences are then translated into clear, natural language descriptions (e.g., "the agent grasps the knife with its right hand").

2. **Forward World Modeling Task.** To create a forward modeling question, we provide the model with the first observation (image) and the complete, ordered sequence of natural language actions. We then present the subsequent observations from the trajectory in a shuffled order. The model's task is to output the correct temporal ordering of these shuffled images. This entire set of information is formatted into a dedicated prompt template.

3. **Inverse World Modeling Task.** For an inverse modeling question, we provide the model with the complete, ordered sequence of observations (images) from the trajectory. We then present the natural language descriptions of the actions in a shuffled order. The model's task is to deduce the correct sequence of actions that connects the given observations. Similarly, this is all inserted into its own prompt template.

4. **Standardized Output Format.** To facilitate robust and automated evaluation, all models are explicitly instructed to provide their answers in the format of a Python list of integers. For instance, if the correct order for three shuffled items (indexed 0, 1, 2) is the second, then the third, then the first, the expected output would be '[1, 2, 0]'. This ensures that responses are unambiguous and easily parsable.

### A.3 DATASET STATISTICS AND EVALUATION DESIGN

Table 5: The 11 predicate classes used to define abstract state changes in our benchmark.

| Predicate Classes | | |
| --- | --- | --- |
| RightGrasping | LeftGrasping | OnTop |
| Inside | Under | Contains |
| Covered | Open | ToggledOn |
| Cooked | Transition | |

### A.3.1 DATASET STATISTICS

Our data generation pipeline, built upon the BEHAVIOR simulator, is designed to be scalable and capable of producing a vast number of distinct QA pairs. For the current iteration of our benchmark,

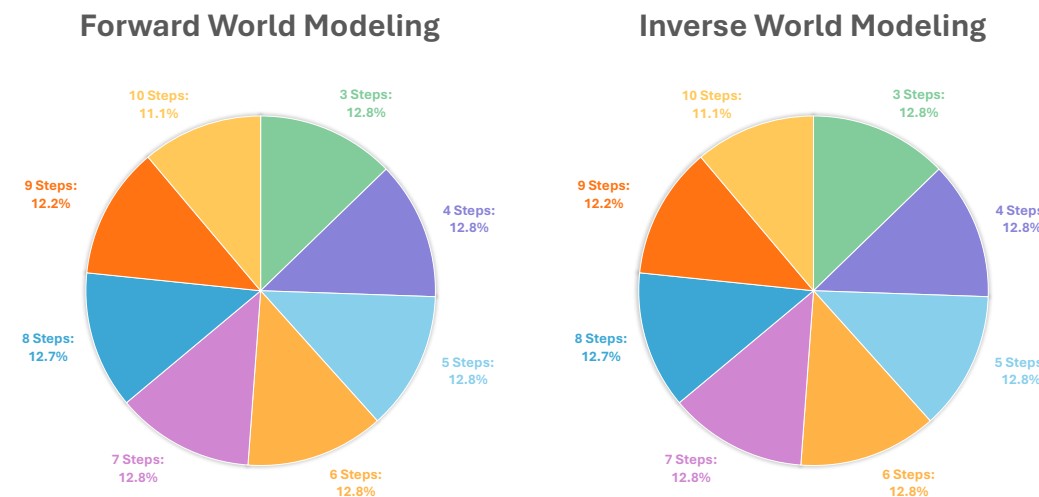

Figure 8: The distribution of problems by the number of steps in our ENACT benchmark dataset is shown for both forward (left) and inverse (right) world modeling tasks. The dataset is balanced, with a nearly uniform distribution of problems ranging from 3 to 10 steps.

we have uniformly sampled a balanced set of 8,972 question-answering pairs from this larger potential pool. This set is evenly divided between the forward and inverse world modeling tasks.

To ensure a comprehensive evaluation of models' reasoning capabilities across different time horizons, the sampled problems feature trajectory lengths varying from 3 to 10 steps. As illustrated in Figure 8, this dataset is intentionally balanced, featuring a near-uniform distribution of problems for each step length across both task types. This balance ensures that our evaluation is not biased towards shorter or longer-term reasoning.

The abstract state changes that define the actions in our benchmark are grounded in a set of 11 symbolic predicates. These predicates describe relationships between the agent and objects, as well as changes in object states. The complete list of predicates is detailed in Table 5.

### A.3.2 EVALUATION DESIGN

**From indices to dynamics.** We grade *what changes*, not just *which index*. Each adjacent state pair yields an *action signature* $a^{\mathrm{sig}}(s_{i-1}, s_i) = \{c = (\gamma, e, \rho)\}$, turning scene-graph deltas into compact semantics (operation $\gamma$ on entity $e$ and predicate $\rho$). For the reference sequence, we compute (i) the *visible* subset $C_i$ and (ii) the *full* set $F_i$. For a prediction, we compute $\tilde{C}_i$ (full diff). This uses state differences as the model's proxy answer and avoids brittle numeric matching.

**Online verifier.** *Forward dynamics.* After reconstructing the shuffled storyboard, we compare the ground-truth index sequence $\tau$ and the prediction $\sigma$. Exact acceptance: $\sigma = \tau$. Semantic acceptance (when lengths match): for all steps $i$,

$$C_i \subseteq \tilde{C}_i.$$

Intuition: the predicted step must *cover* the reference's visible change. The overall decision is `match` = (exact OR semantic); length-mismatched predictions are not accepted (but still get pairwise credit below).

*Inverse dynamics.* The model orders actions. Exact acceptance: indices match. Semantic acceptance (equal length): for all $i$,

$$\tilde{C}_i \subseteq F_i,$$

i.e., the predicted action description can be a concise *subset* of the full reference transition at that position. Again, `match = (exact` OR `semantic)`.

**Metrics.** **Task accuracy (TA).** Score 1 iff the verifier accepts the full prediction, else 0; average over the split:

$$\text{TA} = \frac{1}{|\mathcal{D}|} \sum_{x \in \mathcal{D}} \mathbf{1}\{\text{accepted}(x)\}.$$

**Pairwise accuracy (PA).** Measures stepwise consistency. If lengths match,

$$\text{PA}(x) = \frac{1}{L} \sum_{i=1}^{L} \mathbf{1}\{ C_i \subseteq \tilde{C}_i \text{ (forward) or } \tilde{C}_i \subseteq F_i \text{ (inverse)} \}.$$

Accepted predictions have $\text{PA}(x) = 1$. If lengths differ, we compute PA via a monotone alignment between reference and predicted steps that maximizes the number of subset-satisfying pairs (forward/inverse rule as above). We report the micro-average:

$$\text{PA} = \frac{\sum_x \#\text{correct pairs in } x}{\sum_x L_x}.$$

**Summary.** Multiple valid answers are allowed via the subset rules: forward requires reference-visible $\subseteq$ predicted, inverse requires predicted $\subseteq$ reference-full. TA captures all-or-nothing acceptance; PA gives graded credit for near-correct dynamics.

### A.3.3 ENACT EXAMPLES

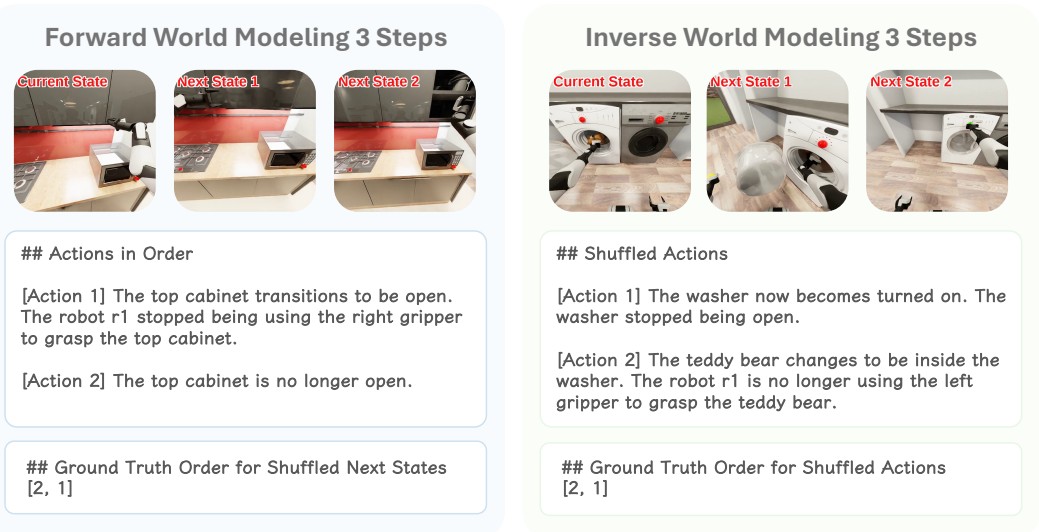

Figure 9: 3-Step **Forward World Modeling** (left) and **Inverse World Modeling** (right) samples.

## B EXPERIMENTS AND ANALYSIS

### B.1 HUMAN ANNOTATION

#### B.1.1 ANNOTATION INTERFACE & HUMAN PERFORMANCE EVALUATION

To establish an empirical upper bound on performance for the ENACT benchmark, we recruited three trained annotators to complete the same set of tasks assigned to the Vision-Language Models (VLMs).

**Forward World Modeling 6 Steps**

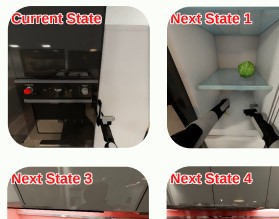
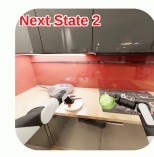
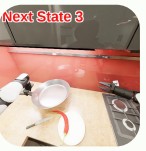
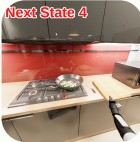
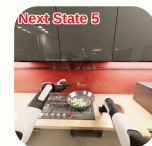

## Actions in Order

[Action 1] The fridge becomes open.

[Action 2] The cutting board now becomes covered by the diced steak. The steak now becomes the diced steak.

[Action 3] The drop-in sink changes to be containing the diced pineapple. The robot r1 changes to be using the right gripper to grasp the carving knife. The cutting board changes to be covered by the diced pineapple. The carving knife is no longer on top of and touching the countertop. The pineapple now becomes the diced pineapple.

[Action 4] The carving knife now becomes on top of and touching the countertop. The robot r1 is no longer using the right gripper to grasp the carving knife.

[Action 5] The robot r1 transitions to be using the right gripper to grasp the bowl. The drop-in sink stopped being containing the diced pineapple. The cutting board stopped being covered by the diced pineapple. The bowl stopped being inside the drop-in sink.

## Ground Truth Order for Shuffled Next States
[3, 4, 1, 5, 2]

**Inverse World Modeling 6 Steps**

## Shuffled Actions

[Action 1] The robot r1 becomes using the right gripper to grasp the head cabbage.

[Action 2] The robot r1 becomes using the left gripper to grasp the frying pan. The robot r1 stopped being using the left gripper to grasp the chili. The frying pan stopped being on top of and touching the countertop.

[Action 3] The robot r1 transitions to be using the right gripper to grasp the carving knife. The robot r1 stopped being using the right gripper to grasp the cutting board. The carving knife is no longer on top of and touching the countertop.

[Action 4] The fridge changes to be open. The robot r1 is no longer using the right gripper to grasp the fridge.

[Action 5] The frying pan now becomes on top of and touching the burner. The frying pan changes to be containing the diced head cabbage. The frying pan becomes covered by the diced head cabbage. The robot r1 is no longer using the left gripper to grasp the frying pan.

## Ground Truth Order for Shuffled Actions
[4, 1, 2, 5, 3]

Figure 10: 6-Step **Forward World Modeling** (left) and **Inverse World Modeling** (right) samples.

**Forward World Modeling 9 Steps**

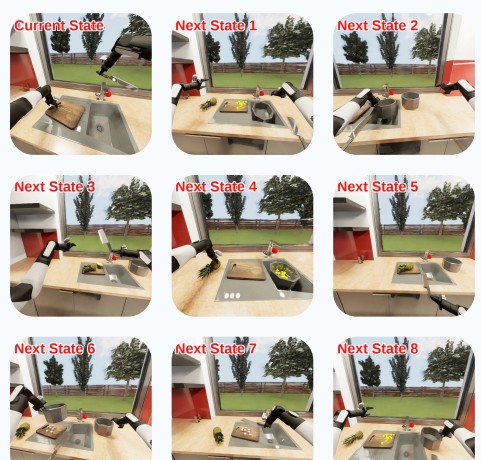

**Inverse World Modeling 9 Steps**

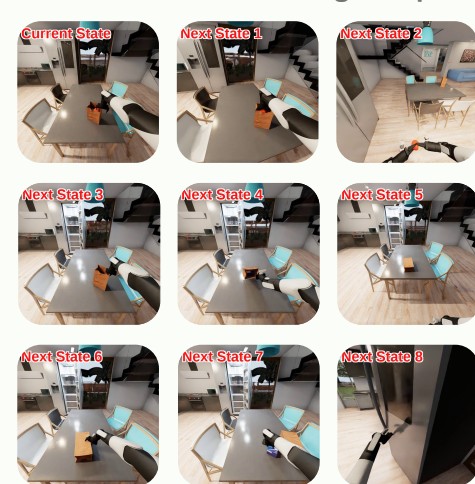

## Actions in Order

[Action 1] The cutting board becomes covered by the diced steak.

[Action 2] The carving knife now becomes on top of and touching the countertop. The robot r1 is no longer using the right gripper to grasp the carving knife.

[Action 3] The bowl transitions to be on top of and touching the drop-in sink.

[Action 4] The robot r1 changes to be using the right gripper to grasp the carving knife. The carving knife stopped being on top of and touching the countertop.

[Action 5] The cutting board is no longer under the pineapple. The pineapple now becomes the half pineapple.

[Action 6] The robot r1 transitions to be using the right gripper to grasp the bowl. The cutting board transitions to be covered by the diced pineapple. The robot r1 is no longer using the right gripper to grasp the carving knife.

[Action 7] The bowl transitions to be inside the drop-in sink. The robot r1 stopped being using the right gripper to grasp the bowl. The bowl stopped being on top of and touching the countertop.

[Action 8] The robot r1 becomes using the left gripper to grasp the half pineapple. The bowl now becomes containing the diced pineapple. The cutting board stopped being covered by the diced pineapple.

## Ground Truth Order for Shuffled Next States
[7, 6, 2, 5, 3, 8, 1, 4]

## Shuffled Actions

[Action 1] The robot r1 now becomes using the right gripper to grasp the beefsteak tomato.

[Action 2] The robot r1 now becomes using the left gripper to grasp the beefsteak tomato. The beefsteak tomato is no longer inside the paper bag.

[Action 3] The fridge is no longer open.

[Action 4] The paper bag changes to be on top of and touching the breakfast table. The robot r1 is no longer using the right gripper to grasp the paper bag.

[Action 5] The paper bag is no longer on top of and touching the breakfast table.

[Action 6] The robot r1 becomes using the right gripper to grasp the paper bag.

[Action 7] The robot r1 is no longer using the right gripper to grasp the paper bag.

[Action 8] The fridge changes to be open. The robot r1 changes to be using the right gripper to grasp the paper bag. The beefsteak tomato becomes inside the fridge. The robot r1 is no longer using the right gripper to grasp the beefsteak tomato. The robot r1 is no longer using the left gripper to grasp the beefsteak tomato.

## Ground Truth Order for Shuffled Actions
[1, 2, 8, 5, 4, 6, 7, 3]

Figure 11: 9-Step **Forward World Modeling** (left) and **Inverse World Modeling** (right) samples.

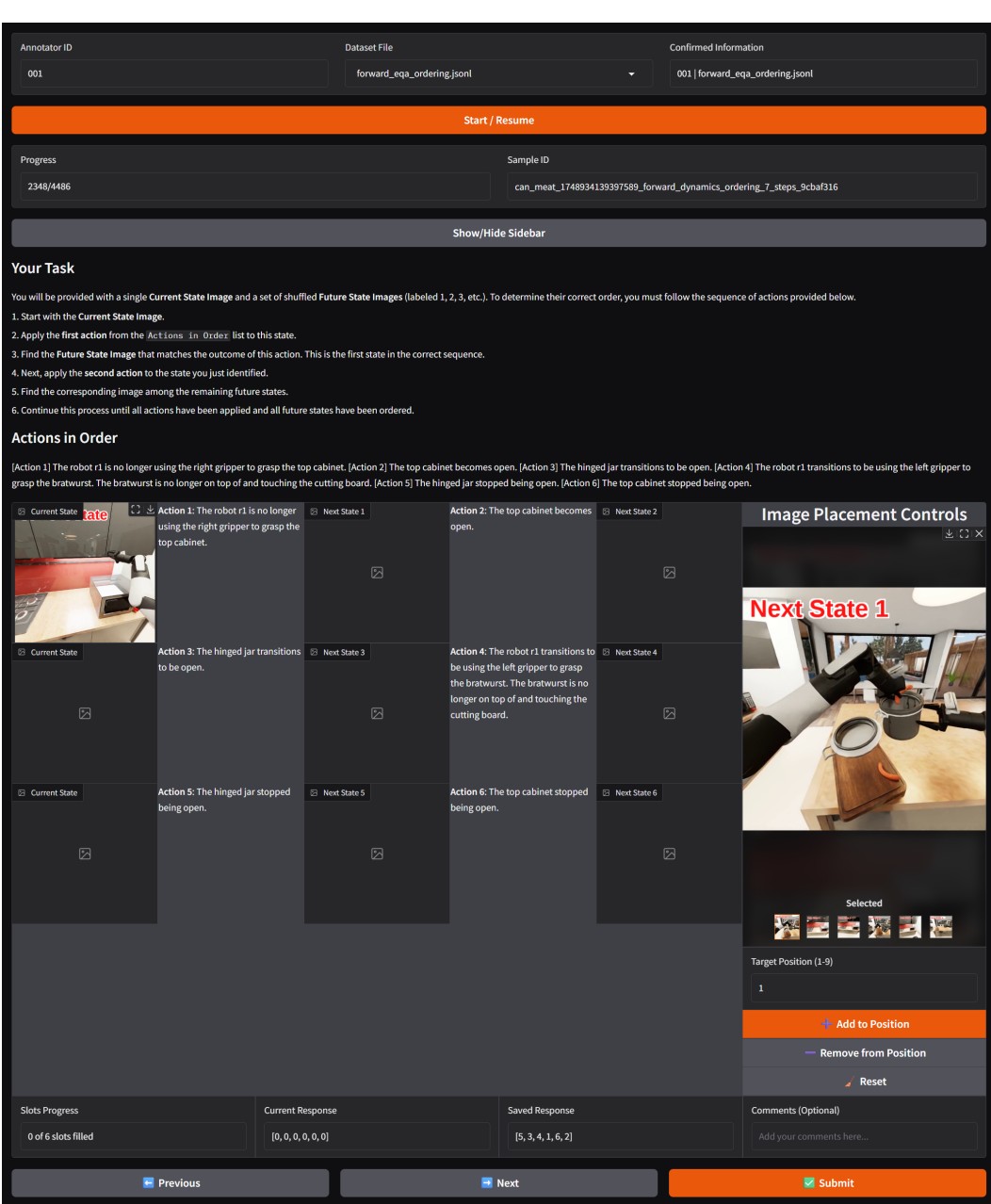

Figure 12: The annotation interface used for evaluating human performance on **Forward World Modeling** problems. Annotators are presented with a *"Current State"* image (top left) and an ordered list of textual actions. The main task is to fill the *"Next State"* slots by selecting the correct image from the shuffled *Candidate Image Library* on the right. The annotator must follow the sequence of actions, using the result of the previous action as the starting point for the next, to determine the correct chronological order of all future states.

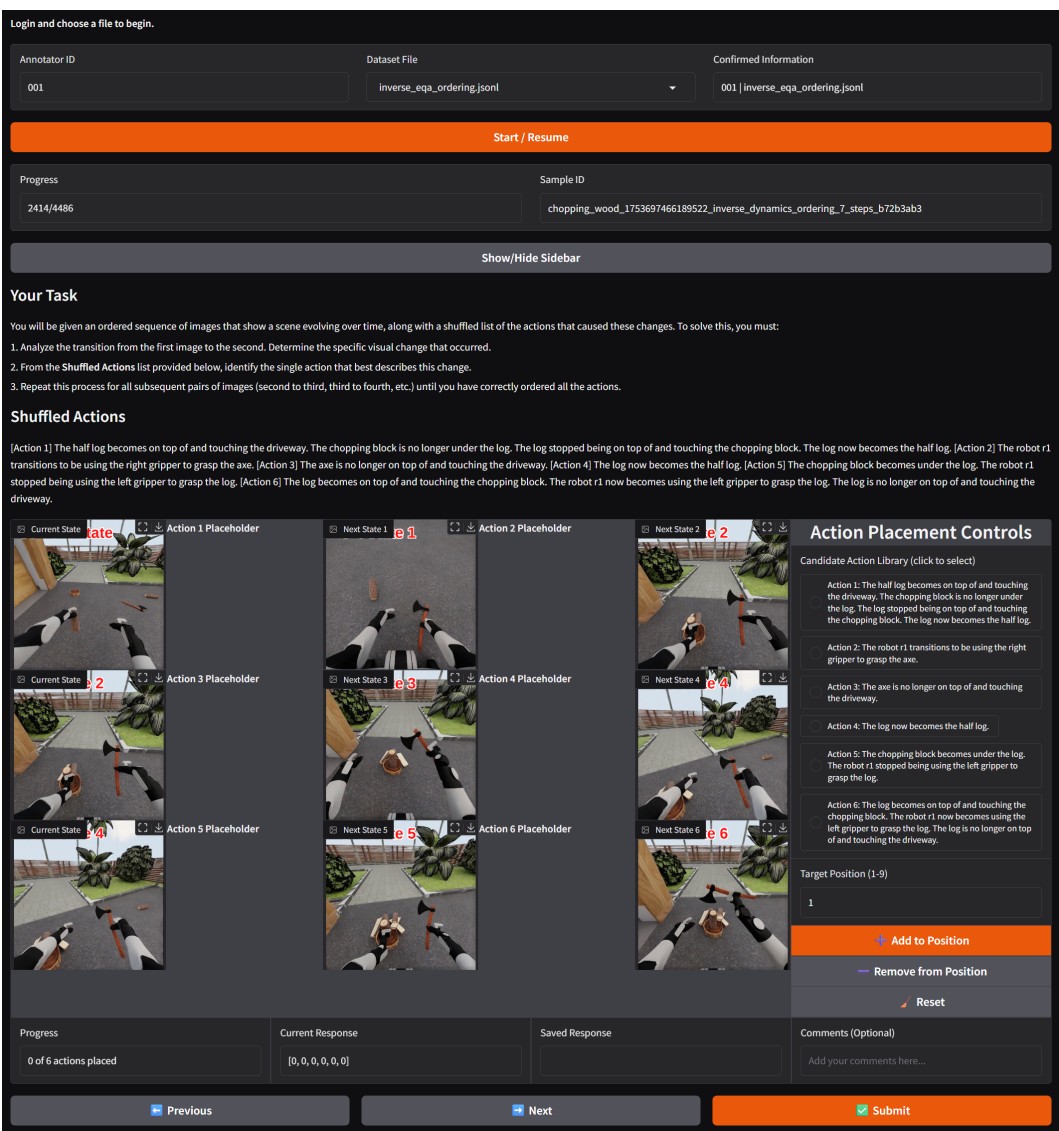

Figure 13: The annotation interface used for evaluating human performance on **Inverse World Modeling** problems. Annotators are shown an ordered sequence of state transitions, displayed as pairs of *"Current State"* and *"Next State"* images. For each transition, their task is to select the correct action description from a shuffled *Candidate Action Library* that caused the visual change between the two states.

The annotators interacted with our customized human annotation interfaces implemented in Gradio, which are illustrated in Figure 12 for the **Forward World Modeling** tasks and in Figure 13 for the **Inverse World Modeling** tasks. Importantly, annotators followed exactly the same instructions and task prompts as those provided to the VLMs, ensuring a fair and consistent comparison between human and model performance. This setup allows us to quantify the extent to which current VLMs approach human-level competence on the ENACT benchmark.

### B.1.2 INTER-ANNOTATOR AGREEMENT ANALYSIS

To ensure the reliability of our human-generated labels, we conducted a rigorous Inter-Annotator Agreement (IAA) analysis. The initial dataset was annotated by three annotators working on disjoint, non-overlapping subsets, which precluded direct agreement measurement. We therefore implemented a systematic cross-annotation protocol. For each task type (forward and inverse) and for each of the eight step-length categories (from 3 to 10 steps), we randomly sampled five questions from each annotator's original assignment. This process created a balanced IAA evaluation set totaling 240 unique questions. Each of these sampled questions was then reassigned to the two annotators who had not performed the original annotation. For example, the five items sampled from Annotator A's work for a given category were re-annotated independently by Annotator B and Annotator C.

Following this protocol, we assessed the resulting annotations using **Krippendorff's Alpha** ($\alpha$) (Krippendorff, 2011), a robust statistical measure that is well-suited for this analysis as it accommodates multiple annotators and is resilient to missing data. Given that our annotation task involved ordering, we configured the analysis for ordinal data. The alpha coefficient is calculated based on the observed and expected disagreement among annotators, according to the formula:

$$\alpha = 1 - \frac{D_o}{D_e}$$

Here, $D_o$ represents the **observed disagreement**, which is calculated from the pairwise differences between all annotations for each item. $D_e$ is the **expected disagreement**, which represents the disagreement that would occur by chance, derived from the marginal distribution of all annotations. This balanced design ensured that every question in the IAA set received three independent labels, allowing for robust pairwise agreement calculation across all three pairs of annotators (A vs. B, A vs. C, and B vs. C) for each condition. To assess the stability of our $\alpha$ coefficient, we computed **95% confidence intervals (CI)** using the bootstrap percentile method with 1,000 resamples of the 240 evaluation items.

Our analysis yielded an overall Krippendorff's Alpha of $\alpha = 0.8320$, with a 95% bootstrap confidence interval of **[0.7879, 0.8682]**. Given the established standard (Krippendorff, 1999; Hayes & Krippendorff, 2007; De Swert, 2012), an alpha value above 0.80 indicates a high level of reliability. This strong result confirms that our annotation guidelines are clear and consistently applied by the annotators.

Pairwise agreement scores were also consistently high, further validating the reliability between individual annotators:

- **Annotator A01 - B02**: $\alpha = 0.8180$
- **Annotator A01 - C03**: $\alpha = 0.8265$
- **Annotator B02 - C03**: $\alpha = 0.8518$

In addition to the chance-corrected alpha metric, we found that the annotators were in perfect agreement on 184 of the 240 selected questions, resulting in a **agreement rate of 76.67%**. Collectively, these strong agreement metrics validate the reliability of our annotation process and the high quality of the resulting dataset.

### B.2 WORLD MODELING AS A PROXY FOR EVALUATING EMBODIED COGNITION

### B.2.1 EXPERIMENTAL SETUP

To ensure a fair and consistent comparison across all models, we employed a standardized evaluation protocol. For each task type (forward and inverse world modeling), a unified question prompt template was used. All input images were resized to a uniform resolution of $512 \times 512$ pixels before being passed to the models. To ensure deterministic and reproducible outputs, the decoding temperature for all models was set to 0. Models were instructed to return their answers as a parsable Python list representing the permutation of indices, as shown in Figure 14 and Figure 15. A comprehensive list of the specific models used in our evaluation is provided in Table 6. We deliberately choose one prompt template across all experiments because we follow the design choice mentioned in Liang et al. (2022b), that the models should adapt to users' input, instead of the reverse case.

Table 6: Details of Vision Language Models (VLMs) assessed in this study.

| Organization | Model Name | Release Date | Full Name | Evaluation Pipeline |
|---|---|---|---|---|
| *Proprietary Models* | | | | |
| OpenAI | GPT-5 | 2025-08 | gpt-5-2025-08-07 | OpenAI API |
| | GPT-5-mini | 2025-08 | gpt-5-mini-2025-08-07 | OpenAI API |
| | GPT-5-nano | 2025-08 | gpt-5-nano-2025-08-07 | OpenAI API |
| Google | Gemini 2.5 Pro | 2025-06 | gemini-2.5-pro | Gemini API |
| | Gemini 2.5 Flash | 2025-06 | gemini-2.5-flash | Gemini API |
| | Gemini 2.5 Flash-Lite | 2025-06 | gemini-2.5-flash-lite | Gemini API |
| Anthropic | Claude Sonnet 4 | 2025-05 | claude-sonnet-4-20250514 | Anthropic API |
| *Open-Weight Models* | | | | |
| Zhipu AI | GLM-4.5V | 2025-08 | GLM-4.5V | Zhipu Foundation Model Open Platform API |
| | GLM-4.1V-Thinking | 2025-07 | GLM-4.1V-Thinking-FlashX | Zhipu Foundation Model Open Platform API |
| Meta | Llama-4-Scout-17B-16E-Ins | 2025-04 | meta-llama/Llama-4-Scout-17B-16E-Instruct | ModelScope API |
| | Llama-4-Mav-17B-128E-Ins | 2025-04 | meta-llama/Llama-4-Mav-17B-128E-Instruct | ModelScope API |
| Shanghai AI Lab | InternVL3.5-241B-A28B | 2025-08 | OpenGVLab/InternVL3.5-241B-A28B | Intern API |
| | InternVL3.5-14B | 2025-08 | OpenGVLab/InternVL3.5-14B | Hugging Face Transformers |
| | InternVL3.5-8B | 2025-08 | OpenGVLab/InternVL3.5-8B | Hugging Face Transformers |
| | InternVL3.5-4B | 2025-08 | OpenGVLab/InternVL3.5-4B | Hugging Face Transformers |
| Google | Gemma-3-27b-it | 2025-03 | google/gemma-3-27b-it | Gemini API |
| | Gemma-3-12b-it | 2025-03 | google/gemma-3-12b-it | Gemini API |
| | Gemma-3-4b-it | 2025-03 | google/gemma-3-4b-it | Gemini API |
| Alibaba | QVQ-72B-Preview | 2024-12 | Qwen/QVQ-72B-Preview | ModelScope API |
| | Qwen2.5-VL-72B-Ins | 2025-01 | Qwen/Qwen2.5-VL-72B-Instruct | ModelScope API |
| | Qwen2.5-VL-32B-Ins | 2025-01 | Qwen/Qwen2.5-VL-32B-Instruct | ModelScope API |
| | Qwen2.5-VL-7B-Ins | 2025-01 | Qwen/Qwen2.5-VL-7B-Instruct | Hugging Face Transformers |
| | Qwen2.5-VL-3B-Ins | 2025-01 | Qwen/Qwen2.5-VL-3B-Instruct | Hugging Face Transformers |
| AIDC | Ovis2.5-9B | 2025-08 | AIDC-AI/Ovis2.5-9B | Hugging Face Transformers |
| | Ovis2.5-2B | 2025-08 | AIDC-AI/Ovis2.5-2B | Hugging Face Transformers |
| OpenBMB | MiniCPM-V-4.5 | 2025-08 | openbmb/MiniCPM-V-4.5 | Hugging Face Transformers |
| | MiniCPM-o-2.6 | 2025-01 | openbmb/MiniCPM-o-2.6 | Hugging Face Transformers |
| Hugging Face | Idefics3-8B-Llama3 | 2024-08 | HuggingFaceM4/Idefics3-8B-Llama3 | Hugging Face Transformers |
| Nvidia | Cosmos-Reason1 | 2025-05 | nvidia/Cosmos-Reason1 | Hugging Face Transformers |

---

### Forward World Modeling Prompt

```
You are a capable agent designed to infer multi-step forward dynamics transitions in
embodied decision-making. Your goal is to predict the correct sequence of future
states that result from applying a given series of actions to an initial state.

## Your Task
You will be provided with a single **Current State Image** and a set of shuffled
**Future State Images** (labeled 1, 2, 3, etc.). To determine their correct order,
you must follow the sequence of actions provided below.

1.  Start with the **Current State Image**.
2.  Apply the **first action** from the 'Actions in Order' list to this state.
3.  Find the **Future State Image** that matches the outcome of this action.
    This is the first state in the correct sequence.
4.  Next, apply the **second action** to the state you just identified.
5.  Find the corresponding image among the remaining future states.
6.  Continue this process until all actions have been applied and all future states
    have been ordered.

## Output Format
Your response **must be only** a Python list of integers representing the correct
chronological order of the future state image labels. Do not include any other text,
reasoning, or explanation.

**Example:**
If you determine the correct sequence is
'Next State 1' -> 'Next State 3' -> 'Next State 2',
Your output must be: `[1, 3, 2]`

## Actions in Order
```

```
{STATE_CHANGES}

Now, please provide your answer in the requested format.
```

Figure 14: The prompt used to evaluate VLMs on the multi-step **Forward World Modeling** task. The model must order shuffled future state images by reasoning over a given action sequence.

### Inverse World Modeling Prompt

```
You are a capable agent designed to infer multi-step inverse dynamics transitions in
embodied decision-making. Your goal is to determine the correct chronological order
of actions that caused the state transitions shown in a sequence of images.

## Your Task
You will be given an ordered sequence of images that show a scene evolving over time,
along with a shuffled list of the actions that caused these changes.

To solve this, you must:
1.  Analyze the transition from the first image to the second. Determine the specific
    visual change that occurred.
2.  From the **Shuffled Actions** list provided below, identify the single action that
    best describes this change.
3.  Repeat this process for all subsequent pairs of images (second to third, third to
    fourth, etc.) until you have correctly ordered all the actions.

## Output Format
Your response **must be only** a Python list of integers representing
the correct order of the action labels.
Do not include any other text, reasoning, explanations, or code formatting.

**Example:**
If the correct sequence is [Action 2] -> [Action 3] -> [Action 1],
your output must be: '[2, 3, 1]'

## Shuffled Actions
{SHUFFLED_ACTIONS}

Now, please provide your answer in the requested format.
```

Figure 15: The prompt used to evaluate VLMs on the multi-step **Inverse World Modeling** task. The model must order a set of shuffled actions by reasoning over an ordered sequence of state images.

### B.2.2 DETAILED ANAYLSIS

A detailed examination of the full experimental results, presented in Table 7 (Task Accuracy) and Table 8 (Pairwise Accuracy), reveals several consistent trends across all evaluated Vision-Language Models (VLMs). We highlight three primary observations below.

First, models consistently achieve higher accuracy on the inverse world modeling task compared to the forward world modeling task. This performance gap is evident across nearly all models, regardless of size or architecture. For instance, in Task Accuracy, Gemini 2.5 Pro scores 81.99% on 3-step forward modeling but 87.76% on 3-step inverse modeling. More notably, this margin widens as the reasoning horizon $L$ increases. For the same model at 10 steps, the forward accuracy drops to 3.60%, while the inverse accuracy remains significantly higher at 14.40%. This suggests that ordering a sequence of known actions to match a visual outcome (inverse) is an easier cognitive task for current VLMs than predicting a sequence of visual outcomes from a set of abstract actions (forward), which requires a more robust capacity for simulation and prediction.

Second, the performance of all VLMs steadily declines as the step length $L$ increases. This trend is universal across both tasks and metrics. The accuracy degradation is particularly sharp for longer horizons, with most models seeing their Task Accuracy fall to near-zero levels for tasks involving 7 or more steps. Even the top-performing proprietary models, such as GPT-5 and Gemini 2.5 Pro, experience a dramatic drop-off, indicating a fundamental challenge in maintaining long-term causal chains and handling the combinatorial complexity that arises with each additional step. This highlights the current limitations of VLMs in multi-step, long-horizon reasoning.

Finally, there is a substantial performance gap between all VLMs and human evaluators. As shown in the final row of each table, human performance is consistently strong and stable across all step lengths, achieving Task Accuracy scores around 90% and Pairwise Accuracy scores exceeding 95%

in many cases. Unlike the models, human accuracy does not degrade sharply as $L$ increases. This near-ceiling performance validates our benchmark as a solvable yet challenging test of embodied cognition and underscores the significant progress required for VLMs to achieve human-level causal understanding and world modeling.

**Why VLMs Perform Better on Inverse than Forward?** The superior performance on inverse modeling can be attributed to the nature of the reasoning involved. The inverse task is a constrained matching problem: the model observes a sequence of visual outcomes and must simply order the corresponding textual actions. This leverages the core strength of VLMs in visual perception and language grounding. Conversely, the forward task is an unconstrained simulation problem. It requires the model to predict a sequence of visual states from abstract actions, demanding a generative understanding of causal dynamics and intuitive physics. This form of predictive world modeling is a known weakness of current VLMs (Tung et al., 2023), which are trained primarily as descriptive, not simulative, systems.

Table 7: **Evaluation on ENACT (Task Accuracy).** Dark gray indicates the best result within each category (Proprietary or Open-Weight Models), and Light gray denotes the second-best result within the category.

| Model | Forward World Modeling | | | | | | | | Inverse World Modeling | | | | | | | |
|---|---|---|---|---|---|---|---|---|---|---|---|---|---|---|---|---|
| | 3 | 4 | 5 | 6 | 7 | 8 | 9 | 10 | 3 | 4 | 5 | 6 | 7 | 8 | 9 | 10 |
| *Proprietary Models* | | | | | | | | | | | | | | | | |
| GPT-5 | 80.59 | 62.72 | 47.13 | 33.62 | 20.24 | 11.58 | 7.30 | 5.00 | 86.19 | 72.65 | 59.65 | 43.73 | 33.68 | 24.04 | 17.15 | 13.00 |
| GPT-5 mini | 83.39 | 62.72 | 45.22 | 31.71 | 19.02 | 9.12 | 5.29 | 2.80 | 84.79 | 67.42 | 58.09 | 41.11 | 29.67 | 18.07 | 13.50 | 8.60 |
| GPT-5 nano | 58.57 | 30.66 | 9.74 | 3.83 | 1.40 | 0.00 | 0.00 | 0.00 | 72.03 | 39.02 | 17.22 | 8.19 | 3.14 | 1.05 | 0.36 | 0.00 |
| Gemini 2.5 Pro | 81.99 | 62.72 | 47.30 | 29.79 | 17.80 | 10.00 | 3.28 | 3.60 | 87.76 | 73.52 | 58.61 | 43.38 | 33.51 | 23.68 | 15.88 | 14.40 |
| Gemini 2.5 Flash | 75.52 | 50.52 | 25.22 | 14.29 | 6.28 | 2.98 | 1.28 | 0.20 | 82.52 | 61.15 | 38.96 | 27.70 | 17.98 | 13.86 | 6.20 | 3.80 |
| Gemini 2.5 Flash-Lite | 52.97 | 27.18 | 10.09 | 3.83 | 1.40 | 0.18 | 0.18 | 0.00 | 69.06 | 42.33 | 19.83 | 8.54 | 4.71 | 0.88 | 0.73 | 0.00 |
| Claude Sonnet 4 | 56.29 | 24.91 | 8.52 | 2.96 | 0.70 | 0.00 | 0.00 | 0.00 | 72.73 | 42.16 | 24.17 | 13.59 | 6.98 | 2.63 | 1.46 | 1.00 |
| *Open-Weight Models* | | | | | | | | | | | | | | | | |
| GLM-4.5V | 66.08 | 40.77 | 18.09 | 8.54 | 1.57 | 0.35 | 0.18 | 0.00 | 79.55 | 57.32 | 32.52 | 20.38 | 11.69 | 5.44 | 1.64 | 0.40 |
| GLM-4.1V-Thinking | 57.52 | 28.40 | 11.30 | 2.26 | 0.35 | 0.18 | 0.00 | 0.00 | 73.43 | 39.37 | 12.00 | 4.53 | 0.87 | 0.53 | 0.00 | 0.00 |
| Llama-4-Scout-17B-16E-Ins | 58.74 | 21.43 | 5.04 | 1.74 | 0.70 | 0.18 | 0.00 | 0.00 | 64.34 | 34.32 | 10.26 | 2.96 | 1.75 | 0.00 | 0.18 | 0.00 |
| Llama-4-Mav-17B-128E-Ins | 63.99 | 32.58 | 14.78 | 4.36 | 1.57 | 0.35 | 0.00 | 0.00 | 71.50 | 49.30 | 24.35 | 11.85 | 4.19 | 1.58 | 0.55 | 0.00 |
| InternVL3.5-241B-A28B | 67.83 | 43.38 | 21.22 | 12.02 | 4.71 | 1.05 | 0.36 | 0.00 | 81.99 | 59.76 | 40.35 | 24.22 | 15.18 | 7.37 | 4.56 | 2.00 |
| InternVL3.5-14B | 46.33 | 14.81 | 3.48 | 1.05 | 0.00 | 0.00 | 0.00 | 0.00 | 66.43 | 45.12 | 23.65 | 11.85 | 5.93 | 1.93 | 1.28 | 0.40 |
| InternVL3.5-8B | 54.72 | 25.09 | 5.39 | 1.05 | 1.22 | 0.18 | 0.00 | 0.00 | 63.99 | 40.24 | 20.00 | 6.79 | 3.49 | 0.53 | 0.36 | 0.20 |
| InternVL3.5-4B | 54.55 | 22.13 | 6.43 | 2.09 | 0.52 | 0.00 | 0.00 | 0.00 | 63.64 | 32.93 | 16.00 | 5.75 | 2.27 | 0.53 | 0.18 | 0.00 |
| Gemma-3-27b-it | 53.15 | 22.82 | 5.57 | 0.87 | 0.17 | 0.18 | 0.00 | 0.00 | 63.46 | 31.88 | 14.61 | 5.05 | 1.57 | 0.35 | 0.00 | 0.60 |
| Gemma-3-12b-it | 51.22 | 21.78 | 6.09 | 1.05 | 0.17 | 0.00 | 0.00 | 0.00 | 52.80 | 27.53 | 9.74 | 2.79 | 1.75 | 0.35 | 0.00 | 0.00 |
| Gemma-3-4b-it | 52.80 | 20.56 | 1.57 | 0.17 | 0.70 | 0.00 | 0.00 | 0.00 | 52.45 | 18.12 | 3.83 | 1.92 | 0.17 | 0.00 | 0.00 | 0.00 |
| QVQ-72B-Preview | 60.84 | 29.79 | 8.17 | 2.09 | 0.70 | 0.00 | 0.00 | 0.00 | 66.96 | 40.24 | 16.87 | 6.97 | 3.84 | 1.23 | 0.55 | 0.00 |
| Qwen2.5-VL-72B-Ins | 71.68 | 40.42 | 18.96 | 7.84 | 3.32 | 1.23 | 0.00 | 0.00 | 75.87 | 53.48 | 29.74 | 17.77 | 11.52 | 4.74 | 1.46 | 0.40 |
| Qwen2.5-VL-32B-Ins | 51.40 | 32.75 | 10.09 | 3.48 | 0.52 | 0.00 | 0.00 | 0.00 | 39.34 | 33.45 | 19.13 | 8.89 | 6.11 | 2.11 | 0.91 | 0.00 |
| Qwen2.5-VL-7B-Ins | 22.73 | 23.17 | 5.39 | 0.52 | 0.17 | 0.00 | 0.00 | 0.00 | 70.10 | 41.11 | 16.52 | 5.23 | 1.05 | 0.00 | 0.00 | 0.00 |
| Qwen2.5-VL-3B-Ins | 45.98 | 13.76 | 5.91 | 0.70 | 0.17 | 0.00 | 0.00 | 0.00 | 56.64 | 32.75 | 13.39 | 5.75 | 1.05 | 0.18 | 0.00 | 0.00 |
| Ovis2.5-9B | 47.55 | 23.00 | 10.61 | 2.96 | 1.05 | 0.18 | 0.00 | 0.00 | 62.76 | 35.54 | 16.00 | 6.27 | 1.75 | 0.35 | 0.00 | 0.00 |
| Ovis2.5-2B | 39.69 | 17.77 | 5.91 | 0.87 | 0.52 | 0.00 | 0.00 | 0.00 | 48.43 | 23.87 | 8.52 | 1.57 | 0.00 | 0.00 | 0.00 | 0.00 |
| MiniCPM-V-4.5 | 48.43 | 19.16 | 8.35 | 1.92 | 0.52 | 0.18 | 0.00 | 0.00 | 68.01 | 37.98 | 22.09 | 9.41 | 3.66 | 1.75 | 0.18 | 0.20 |
| MiniCPM-o-2.6 | 26.05 | 17.07 | 5.22 | 1.57 | 0.00 | 0.00 | 0.00 | 0.00 | 38.64 | 27.35 | 11.30 | 2.44 | 0.52 | 0.18 | 0.00 | 0.00 |
| Idefics3-8B-Llama3 | 48.08 | 16.20 | 2.26 | 0.52 | 0.17 | 0.00 | 0.00 | 0.00 | 46.33 | 16.72 | 2.96 | 1.57 | 0.00 | 0.00 | 0.00 | 0.00 |
| Cosmos-Reason1 | 45.45 | 21.43 | 5.04 | 0.52 | 0.17 | 0.00 | 0.00 | 0.00 | 51.92 | 29.09 | 12.02 | 3.31 | 0.52 | 0.18 | 0.00 | 0.00 |
| **Human Performance** | 90.38 | 92.16 | 89.74 | 85.71 | 88.31 | 87.02 | 85.58 | 84.00 | 91.78 | 90.24 | 88.70 | 88.15 | 89.53 | 92.28 | 87.73 | 85.00 |

## B.3 PROBING EXPERIMENTS COMMON SETUP

To gain deeper insights into model sensitivities, we conducted a series of probing experiments. This section outlines the common experimental framework that applies to our analyses of Image Realism (Section B.4), Camera Configurations (Section B.5), and Robot Appearance (Section B.6.1).

For these experiments, we selected two representative models. Given its strong balance of performance and computational cost in our main results, we chose **GPT-5 mini** as our primary model to represent state-of-the-art proprietary VLMs. To include a strong open-weight counterpart, we also selected **InternVL3.5-241B-A28B**, which demonstrated robust performance among open models.

To ensure the experiments were comprehensive yet manageable, we focused on step lengths $L \in \{3, 6, 9\}$, covering short, medium, and long-term reasoning horizons. For each of these lengths and for both the forward and inverse tasks, we randomly sampled 50 questions. This resulted in a consistent test bed of 300 total question-answering pairs for each experimental setting. We maintained the same question prompts used in the main benchmark experiments to isolate the effect of the variable being tested. The default setting used in our benchmark dataset serves as the baseline for all comparisons.

Table 8: **Evaluation on ENACT (Pairwise Accuracy).** Dark gray indicates the best result within each category (Proprietary or Open-Weight Models), and Light gray denotes the second-best result within the category.

| Model | Forward World Modeling | | | | | | | | Inverse World Modeling | | | | | | | |
|---|---|---|---|---|---|---|---|---|---|---|---|---|---|---|---|---|
| | 3 | 4 | 5 | 6 | 7 | 8 | 9 | 10 | 3 | 4 | 5 | 6 | 7 | 8 | 9 | 10 |
| *Proprietary Models* | | | | | | | | | | | | | | | | |
| GPT-5 | 84.62 | 75.26 | 69.96 | 64.18 | 57.48 | 52.16 | 49.45 | 46.93 | 86.28 | 80.37 | 76.09 | 68.78 | 65.71 | 62.13 | 57.12 | 55.33 |
| GPT-5 mini | 87.50 | 76.25 | 70.65 | 63.41 | 58.14 | 52.38 | 46.65 | 44.11 | 85.05 | 76.77 | 75.43 | 67.67 | 63.79 | 57.04 | 55.04 | 50.02 |
| GPT-5 nano | 67.83 | 50.29 | 38.61 | 30.35 | 25.97 | 21.90 | 17.59 | 16.84 | 72.81 | 53.95 | 42.48 | 36.45 | 31.68 | 28.20 | 24.11 | 20.33 |
| Gemini 2.5 Pro | 86.10 | 76.42 | 69.83 | 60.80 | 53.26 | 48.12 | 40.12 | 36.98 | 87.94 | 81.18 | 75.39 | 70.03 | 66.03 | 62.91 | 57.78 | 56.62 |
| Gemini 2.5 Flash | 81.64 | 67.94 | 54.17 | 43.38 | 37.43 | 32.73 | 29.88 | 28.07 | 82.78 | 72.18 | 60.83 | 58.19 | 53.14 | 51.78 | 47.99 | 44.98 |
| Gemini 2.5 Flash-Lite | 64.34 | 49.07 | 38.70 | 33.87 | 27.81 | 25.44 | 23.31 | 20.31 | 69.58 | 57.55 | 46.04 | 39.09 | 34.06 | 30.18 | 27.51 | 23.16 |
| Claude Sonnet 4 | 65.65 | 45.82 | 36.65 | 30.52 | 26.61 | 22.78 | 21.49 | 20.16 | 73.25 | 56.85 | 48.87 | 43.07 | 37.00 | 32.71 | 30.50 | 28.49 |
| *Open-Weight Models* | | | | | | | | | | | | | | | | |
| GLM-4.5V | 74.30 | 59.99 | 47.65 | 38.78 | 30.83 | 25.69 | 21.60 | 19.67 | 80.59 | 69.28 | 57.04 | 51.53 | 46.95 | 41.68 | 37.36 | 37.93 |
| GLM-4.1V-Thinking | 67.31 | 49.48 | 38.43 | 31.29 | 25.80 | 21.50 | 20.14 | 18.73 | 75.35 | 56.27 | 46.57 | 36.79 | 29.61 | 24.56 | 23.91 | 25.80 |
| Llama-4-Scout-17B-16E-Ins | 68.18 | 42.62 | 34.30 | 30.52 | 28.50 | 26.57 | 25.94 | 31.20 | 66.00 | 50.00 | 41.30 | 37.04 | 29.73 | 25.61 | 22.45 | 26.54 |
| Llama-4-Mav-17B-128E-Ins | 72.47 | 52.09 | 43.87 | 35.30 | 29.90 | 25.89 | 22.79 | 20.49 | 72.55 | 62.60 | 50.52 | 43.10 | 35.17 | 31.68 | 28.10 | 25.80 |
| InternVL3.5-241B-A28B | 75.79 | 62.25 | 50.83 | 45.85 | 37.84 | 32.88 | 27.85 | 25.24 | 82.26 | 70.09 | 60.61 | 53.38 | 45.90 | 39.35 | 34.12 | 30.56 |
| InternVL3.5-14B | 54.90 | 36.53 | 27.87 | 25.47 | 22.02 | 18.73 | 18.29 | 20.60 | 69.06 | 59.52 | 49.00 | 43.45 | 37.61 | 32.28 | 29.31 | 28.58 |
| InternVL3.5-8B | 64.42 | 44.83 | 31.48 | 24.32 | 23.62 | 21.50 | 19.30 | 25.47 | 65.03 | 56.10 | 45.35 | 37.67 | 35.02 | 29.62 | 26.41 | 23.60 |
| InternVL3.5-4B | 63.11 | 42.04 | 30.26 | 26.13 | 21.73 | 20.28 | 19.64 | 21.98 | 64.95 | 50.12 | 41.61 | 35.78 | 29.00 | 26.57 | 27.55 | 24.04 |
| Gemma-3-27b-it | 63.29 | 44.66 | 32.04 | 25.82 | 22.11 | 19.50 | 16.74 | 16.29 | 64.95 | 48.37 | 40.04 | 33.87 | 28.53 | 23.63 | 21.74 | 19.36 |
| Gemma-3-12b-it | 62.33 | 43.55 | 32.78 | 25.68 | 22.45 | 20.40 | 17.70 | 16.71 | 53.23 | 43.79 | 34.43 | 29.90 | 25.57 | 22.31 | 21.60 | 18.16 |
| Gemma-3-4b-it | 61.98 | 41.17 | 35.70 | 35.16 | 30.51 | 26.17 | 26.73 | 25.80 | 53.06 | 36.41 | 29.52 | 26.38 | 22.66 | 24.44 | 33.71 | 33.62 |
| QVQ-72B-Preview | 69.14 | 52.96 | 40.83 | 36.27 | 33.16 | 30.63 | 26.30 | 24.76 | 71.33 | 58.77 | 48.43 | 44.36 | 40.26 | 39.30 | 36.66 | 36.58 |
| Qwen2.5-VL-72B-Ins | 78.15 | 60.05 | 49.87 | 41.92 | 36.77 | 31.73 | 28.03 | 25.07 | 77.80 | 65.85 | 53.30 | 48.19 | 44.07 | 37.57 | 33.76 | 36.27 |
| Qwen2.5-VL-32B-Ins | 67.83 | 55.46 | 44.35 | 35.75 | 27.52 | 26.42 | 22.01 | 18.07 | 63.55 | 59.70 | 54.57 | 51.01 | 49.36 | 47.17 | 41.47 | 40.16 |
| Qwen2.5-VL-7B-Ins | 26.84 | 43.90 | 32.00 | 23.07 | 19.66 | 16.69 | 11.82 | 11.31 | 70.54 | 56.45 | 42.43 | 32.89 | 25.07 | 19.52 | 16.72 | 17.42 |
| Qwen2.5-VL-3B-Ins | 58.22 | 35.31 | 30.57 | 24.08 | 20.36 | 17.44 | 14.87 | 15.07 | 57.43 | 49.13 | 40.48 | 34.88 | 28.33 | 26.14 | 22.97 | 20.51 |
| Ovis2.5-9B | 58.39 | 42.51 | 34.96 | 31.08 | 24.61 | 20.78 | 18.11 | 16.96 | 64.86 | 51.74 | 41.65 | 35.47 | 30.95 | 26.64 | 23.70 | 23.25 |
| Ovis2.5-2B | 46.94 | 38.85 | 32.65 | 26.86 | 25.63 | 22.21 | 22.49 | 24.87 | 54.28 | 44.08 | 35.43 | 29.06 | 27.84 | 25.56 | 27.62 | 29.29 |
| MiniCPM-V-4.5 | 60.75 | 38.73 | 33.65 | 25.47 | 24.81 | 21.40 | 21.60 | 18.33 | 69.23 | 53.08 | 47.35 | 39.55 | 34.87 | 30.63 | 27.05 | 25.71 |
| MiniCPM-o-2.6 | 35.31 | 39.37 | 29.48 | 31.78 | 27.66 | 26.39 | 24.59 | 27.42 | 54.11 | 48.26 | 44.70 | 40.00 | 38.28 | 36.12 | 33.23 | 31.71 |
| Idefics3-8B-Llama3 | 60.23 | 36.99 | 31.83 | 24.25 | 21.29 | 20.80 | 20.46 | 17.71 | 47.38 | 33.86 | 27.26 | 23.48 | 19.87 | 18.50 | 17.04 | 15.16 |
| Cosmos-Reason1 | 56.28 | 41.86 | 34.75 | 28.40 | 26.46 | 26.49 | 25.41 | 24.88 | 58.30 | 45.93 | 44.25 | 38.50 | 35.72 | 34.56 | 31.50 | 28.64 |
| **Human Performance** | 93.62 | 95.30 | 95.04 | 93.87 | 95.43 | 95.41 | 94.75 | 95.13 | 92.05 | 93.56 | 94.35 | 94.25 | 95.96 | 97.74 | 96.30 | 96.29 |

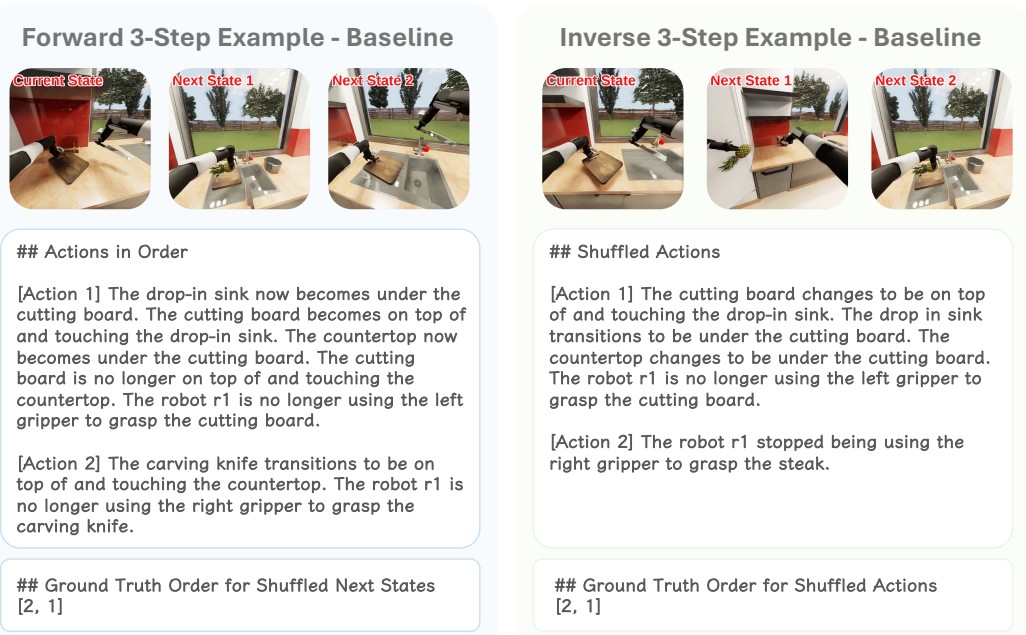

Figure 16: Illustrative trajectories of **Forward World Modeling** and **Inverse World Modeling** for a representative baseline question.

For each setting, we report the **Pairwise Accuracy** along with its standard error. In our summary heatmaps (Figure 4 for GPT-5 mini and Figure 17 for InternVL3.5-241B-A28B), we use $\Delta$ to visualize the performance difference between a variant and the baseline. To assess the statistical significance of these differences, we perform a two-tailed unpaired Welch's t-test. An unpaired test is appropriate as each question is evaluated in an independent session. We specifically use Welch's t-test as it does not assume equal variance between the two groups being compared (baseline vs. variant). We report the p-value for each comparison and consider a result to be statistically significant if $p < 0.05$. We qualitatively classify any performance change where $|\Delta| < 0.05$ as a small change. We show one baseline question and its images for both forward and inverse settings in Figure 16, and for other settings, we *only show their images, as they all share the same question text and answers.*

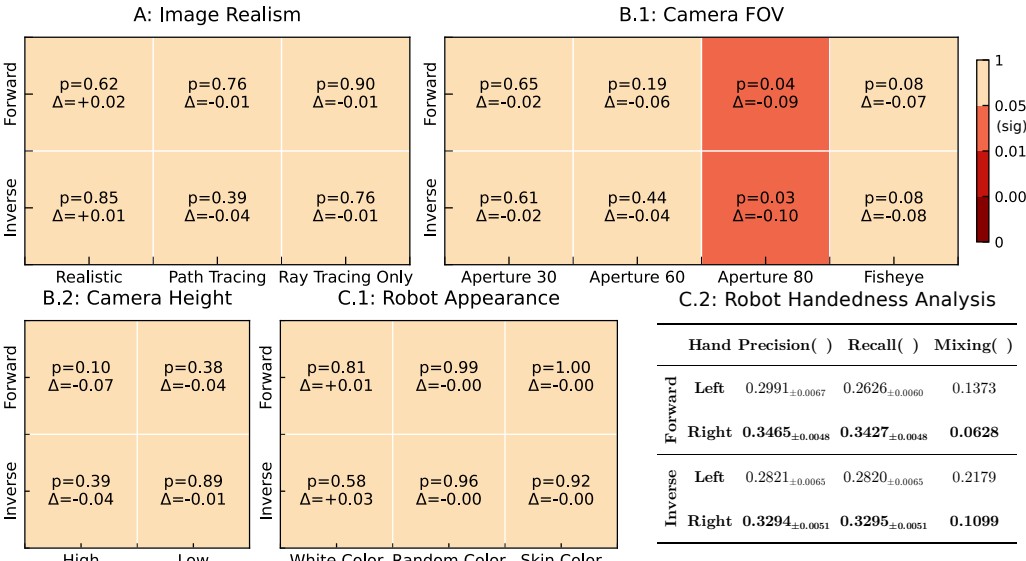

Figure 17: **Probing experiment results with InternVL3.5-241B-A28B on ENACT.** Heatmaps show two-tailed unpaired p-values against the baseline, using *Pairwise Accuracy*. $p < 0.05$ is considered *significant*. Darker red means more significant. $\Delta$ is the performance change from the baseline. If *significant* and $\Delta < 0$, the setting is worse than the baseline. C.2 reports the robot's performance on the left- and right-hand predicates, where *Mixing* is the proportion of ground truth left or right cases that are predicted as the other hand (i.e., mixing one hand into the other hand). Note that, although InternVL3.5-241B-A28B performance is less significant than GPT-5 mini, the $|\Delta|$ across unnatural camera configurations still remains high ($> 0.05$) when the same settings are significant for GPT-5 mini.

## B.4 Sensitivity to Image Realism

Although the BEHAVIOR simulator is designed to be photo-realistic, we were curious whether a "sim-to-real" gap might still exist due to subtle differences in rendering quality. Specifically, we sought to investigate if such a gap affects performance on our world modeling tasks and to quantify the impact of rendering fidelity on the reasoning capabilities of state-of-the-art Vision-Language Models, such as GPT-5 mini. In the following sections, we detail the experimental setup for evaluating model performance across various levels of image realism.

### B.4.1 Realistic: Generated Images as Real World Proxy

Since our activities are diverse and complex, reproducing simulator outputs in the real world on a one-to-one basis would incur prohibitively high costs. However, with the advent of powerful image generation models with the ability of image-scale reproduction (e.g., GPT-image-1), it is feasible to use them as a real-world proxy to convert frames rendered by simulator into realistic styles, which provides a cost-effective and well-aligned alternative.

Constructing prompts for high-accuracy style transfer poses several challenges. First, since our segmented frames are extracted from a replayed robot trajectory, the generated realistic frames

corresponding to the trajectory must preserve consistent content and style, including object shapes and appearances, lighting conditions, material properties, and camera parameters. Second, image generation models often demonstrate instability and errors in understanding fine-grained structures of robotic arms (particularly the gripper) and in interpreting robotic actions. To mitigate these issues, we establish a detailed set of rules and incorporate them into the prompt design (Figure 20), which improves both stability and fidelity in the generated outputs.

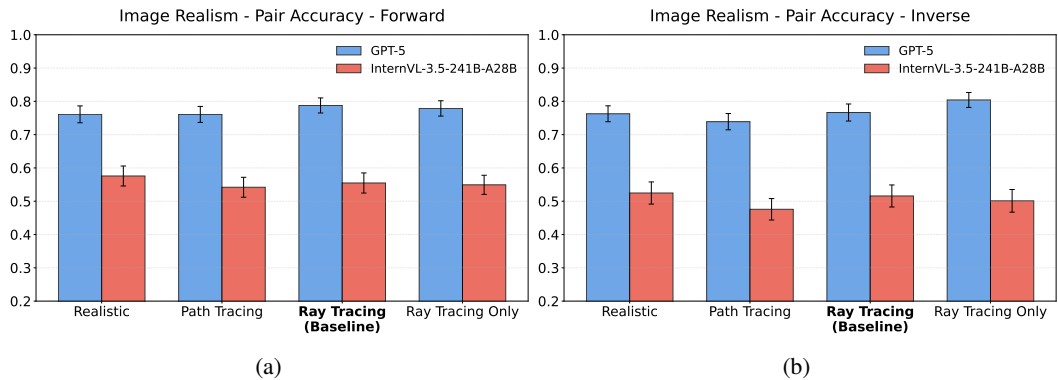

(a)  (b)

Figure 18: **Probing image realism with GPT-5 and InternVL3.5-241B-A28B.** (a) Forward dynamics; (b) Inverse dynamics. Bar plots report Pairwise Accuracy across four rendering settings—Realistic, Path Tracing, Ray Tracing (Baseline), and Ray Tracing Only. Error bars denote ±SEM. The baseline x-tick is bolded.

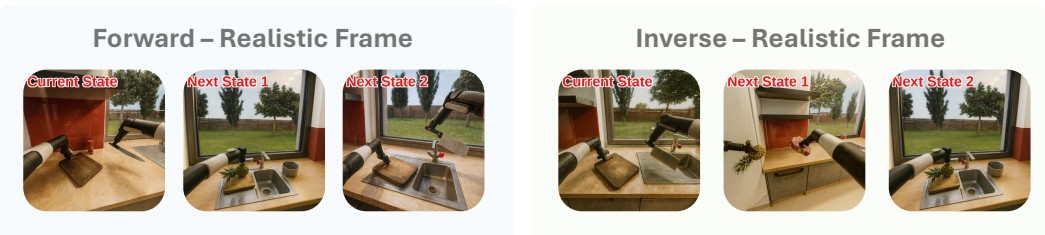

Figure 19: Examples of simulator frames converted into realistic styles for both **Forward World Modeling** (left) and **Inverse World Modeling** (right) trajectories.

```
Project Instruction and Prompt for Image Generation

## Below are the instructions and regulations, treat them as the sole, global reference
for all image generations you are going to perform.

## Core Objective
Convert simulator screenshots into photorealistic PBR images. Change style only;
do not change content.

## Content Lock (Content-Locked)
Preserve the count, position, size, geometry, and pose of all objects.
The robot hand and knife angles, shapes, and actions must match exactly.

## Camera crop and viewpoint must remain unchanged.
The outdoor scene must remain daytime; tree and fence silhouettes must not change.
If realism conflicts with content, content fidelity takes precedence.

## Style Requirements
* Lighting: Warm under-cabinet tungsten (3200-3600 K) + soft window daylight fill.
* Tone: Filmic contrast, smooth highlight roll-off, no crushed blacks or blown
highlights.
* Camera: approximately 35 mm, f/2.8-4, shallow DOF; subject sharp with gently blurred
background.
* Shadows: Realistic soft shadows, contact shadows, and ambient occlusion.
```

```
## Materials:
* Metal knife and trims: Brushed, anisotropic metal.
* Robot: Matte polymer.
* Cutting board and countertop: Sealed/oiled wood grain.
* Glass/walls: Glossy glass with realistic reflections and refractions.
* Post-processing: Subtle camera grain; light vignette.
* Prohibited: Cartoonish look, plastic sheen, bloom, oversaturation, hard outline
sharpening, fake lighting effects.

## Acceptance Criteria
* Edge alignment: SSIM >= 0.95 (along object boundaries).
* Segmentation: IoU >= 0.98 for robot, knife, cutting board, outdoors.
* Color difference: delta Hue <= 3°, delta L <= 6.
* Knife shape error: <= 1 px.
* Outdoor tree/fence silhouette error: <= 1-2 px.

## Implementation Suggestions
* Use low denoise strength 0.20-0.35, CFG 4-6.
* Negative prompt: forbid new objects, geometry changes,
cartoonish/oversaturated/plastic textures.
* Detail pass: add micro-surface material detail + light film grain.

Now, review and summarize what you have learned from these instructions.

Following the instructions you have learned, transform the given image into realistic
photograph style.
```

Figure 20: The prompt used to generate realistic photographic style images from segmented frames (of a replayed robot trajectory).

### B.4.2 PATH TRACING SETUP

To generate the highest-fidelity images for our analysis of image realism, we utilized path tracing. This was achieved directly through the built-in, real-time path tracing engine provided by the NVIDIA Isaac Sim simulator. An example can be seen in Figure 21.

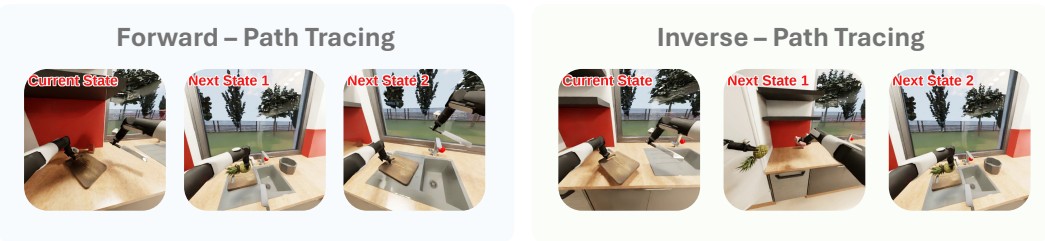

Figure 21: The figure illustrates **Forward World Modeling** (left) and **Inverse World Modeling** (right) trajectories rendered using the path tracing engine in NVIDIA Isaac Sim.

### B.4.3 RAY TRACING ONLY SETUP

This setup was designed to represent an intermediate rendering quality (representing 'unrealistic'). While it still utilizes the ray tracing pipeline as its foundation, we manually disabled several advanced lighting and post-processing effects to reduce visual fidelity. Specifically, we turned off the following features: reflections, DLSS, ambient occlusion, sampled lighting, ambient light, and flow. The resulting visual style, which lacks these richer effects, can be seen in Figure 22.

### B.5 SENSITIVITY TO CAMERA CONFIGURATIONS

### B.5.1 CAMERA APERTURE SETUP

Our default baseline is aperture 40. We also investigate apertures 30, 60, and 80. Examples can refer to Figure 23, 24, and 25.

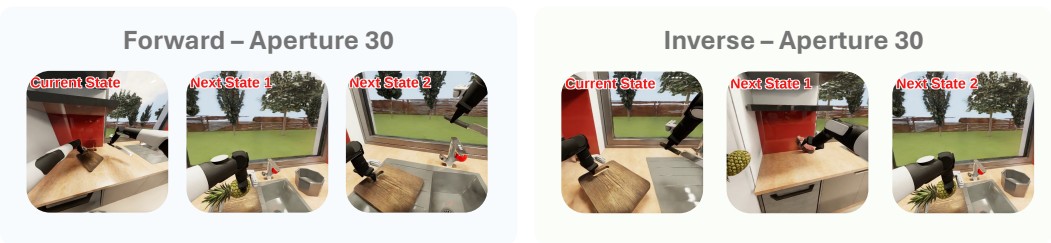

Figure 22: Examples of an intermediate rendering style created with a simplified ray tracing pipeline for **Forward World Modeling** (left) and **Inverse World Modeling** (right) trajectories.

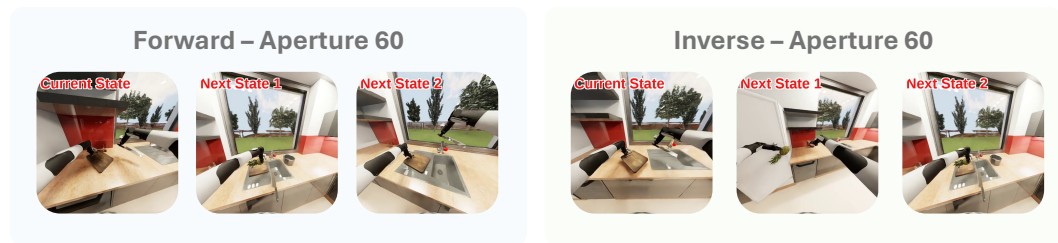

Figure 23: Example trajectories of **Forward World Modeling** (left) and **Inverse World Modeling** (right), captured with a camera aperture of 30.

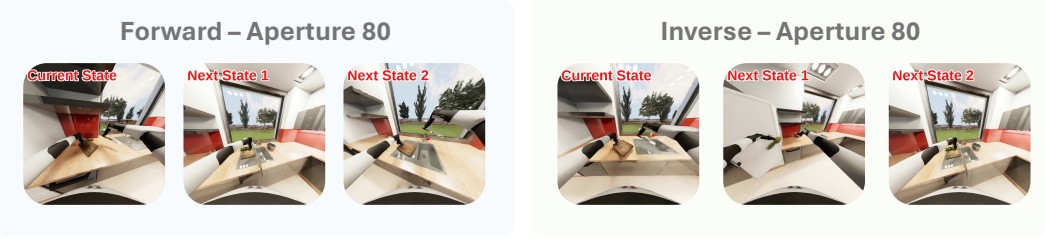

Figure 24: Example trajectories of **Forward World Modeling** (left) and **Inverse World Modeling** (right), captured with a camera aperture of 60.

Figure 25: Example trajectories of **Forward World Modeling** (left) and **Inverse World Modeling** (right), captured with a camera aperture of 80.

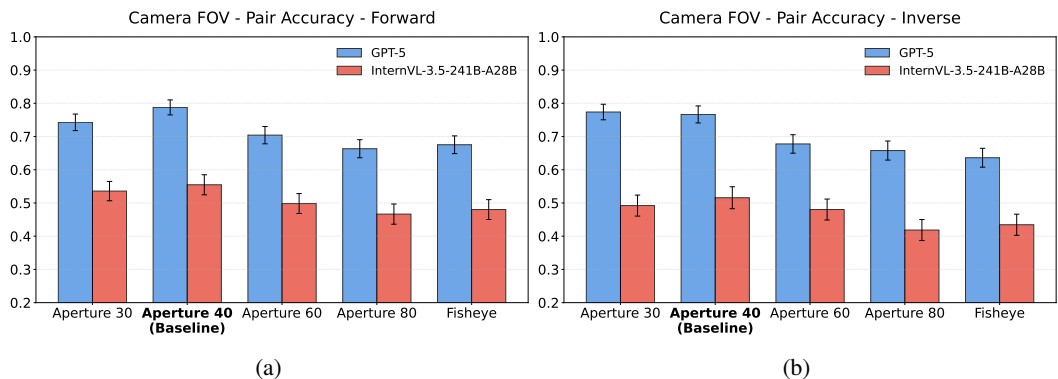

Figure 26: **Probing camera field-of-view (FOV) with GPT-5 and InternVL-3.5-241B-A28B.** (a) Forward dynamics; (b) Inverse dynamics. Bar plots report Pairwise Accuracy across five lens settings—Aperture 30, Aperture 40 (Baseline), Aperture 60, Aperture 80, and Fisheye. Error bars denote ±SEM; the baseline tick is bolded.

### B.5.2 FISHEYE LENS SETUP

Isaac Sim provides the fisheye lens settings. We choose `fisheyePolynomial`, which is the most similar to a daily fisheye lens, such as GoPro, as our evaluated target. The effect can be seen in the example Figure 27.

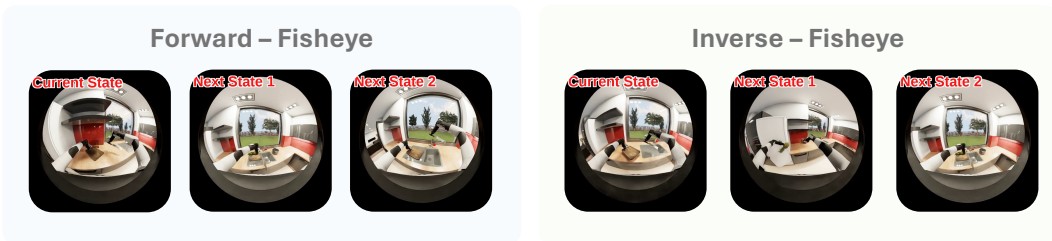

Figure 27: Example trajectories of **Forward World Modeling** (left) and **Inverse World Modeling** (right), captured with a fisheye-style camera.

### B.5.3 CAMERA HEIGHT SETUP

The default setting height is 1.75 m, we also investigate the high (+0.5m) setting and low (−0.25m)setting, and the examples are shown in Figure 28 and 29.

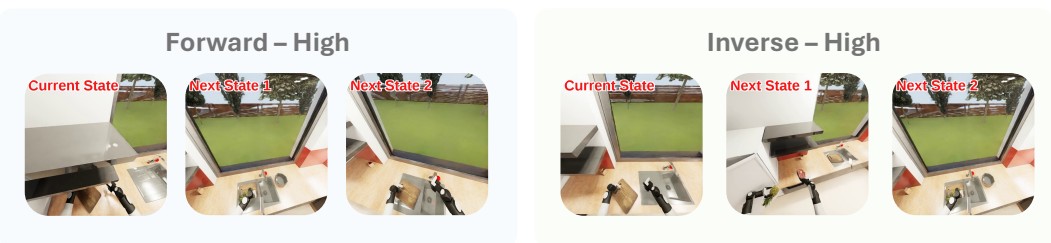

Figure 28: Example trajectories of **Forward World Modeling** (left) and **Inverse World Modeling** (right), captured from a camera height of 2.25 m.

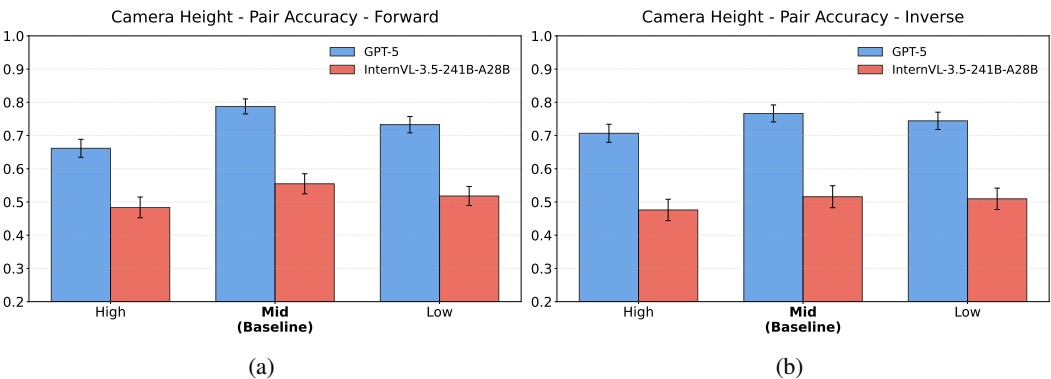

Figure 29: Example trajectories of **Forward World Modeling** (left) and **Inverse World Modeling** (right), captured from a camera height of 1.5 m.

Figure 30: **Probing camera height with GPT-5 and InternVL-3.5-241B-A28B.** (a) Forward dynamics; (b) Inverse dynamics. Bar plots report Pairwise Accuracy across three viewpoints—High, Mid (Baseline), and Low. Error bars denote ±SEM; the baseline tick is bolded.

## B.6 DO VLMS HAVE EMBODIED BIASES?

### B.6.1 ROBOT APPEARANCE

We test three variants: White Color, Random Color (robot color is randomized at each frame), and Skin Color (robot is rendered with a human-like skin tone). Examples can be referred to Figure 31, 32, and 33.

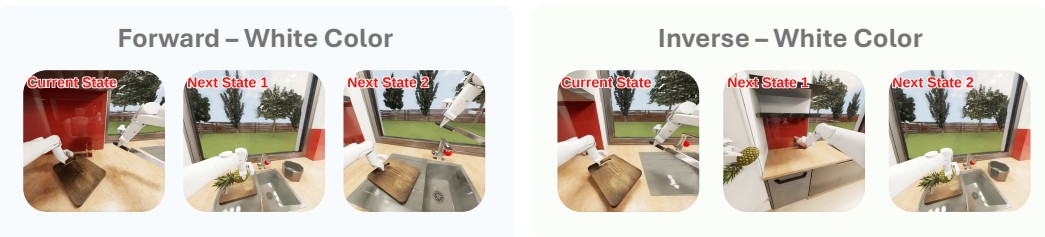

Figure 31: Example trajectories of **Forward World Modeling** (left) and **Inverse World Modeling** (right), with the robot gripper rendered in white.

### B.6.2 HANDEDNESS ASYMMETRY

Based on our experimental setup (C.1). We further examine whether predictions involving agent interactions reflect real-world handedness asymmetry (typically favoring the right hand).

The analysis shows that humans consistently outperformed all models in precision, recall, and hand-mixing rate. Among models, InternVL-3.5-241B-A28B showed the weakest performance, while GPT-5 and GPT-5 mini were more balanced across forward and inverse tasks than Gemini-2.5 Pro, which favored the inverse setting. A consistent pattern also emerges: in both humans and models, and across both task types, right-hand precision and recall systematically exceed those of the left (Figures 36, 37). Furthermore, left-to-right mixing rate (ground-truth left-hand components wrongly

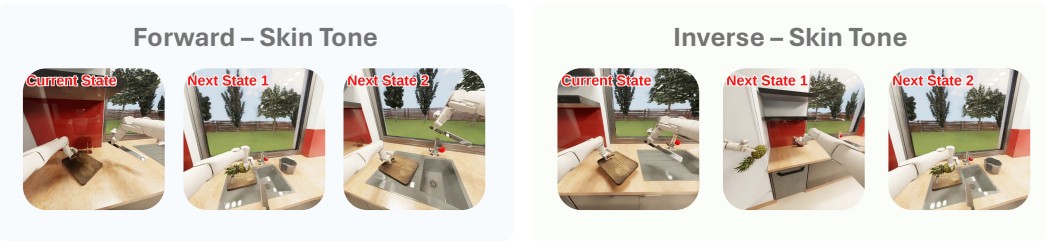

Figure 32: Example trajectories of **Forward World Modeling** (left) and **Inverse World Modeling** (right), with the robot gripper rendered in a random color at each frame.

Figure 33: Example trajectories of **Forward World Modeling** (left) and **Inverse World Modeling** (right), with the robot gripper rendered in a human skin–like color.

predicted as right-hand ones) substantially exceeds the reverse (Figure 38). This analysis also shows a consistent right-hand advantage across forward and inverse tasks, for both models and humans in precision, recall, and hand-mixing rate. Right-hand usage appears to serve as a cognitive and statistical default, which may shaped by real-world dominance and data imbalance, while left-hand recognition appears to be more fragile.

## C  ERROR ANALYSIS

### C.1  METHODOLOGY FOR ERROR CALCULATIONS

**Signature Modeling from Scene Graph-level Differences**   For error analysis, it is found hard to recognize predicate-level or semantic-level errors through natural language-based actions, i.e. visible differences between consecutive states. Hence, we parse the raw (natural language) action predicates as a signature into a sequence of unique state-change signature $(a_0^{\mathrm{sig}}, a_1^{sig}, \dots)$.

$$c_i ::= (\gamma, e_1, \rho, e_2) \mid (\gamma, e, \rho) \mid (\mathrm{transition}, e, \rho_{\mathrm{from}} \to \rho_{\mathrm{to}}), \quad \gamma \in \{\mathrm{add, remove}\}$$

To further structure these signatures, each signature $a_i^{\mathrm{sig}}$ is then modeled as a finite set of components $\{c_1, c_2, \dots\}$. Each component $c_i$ represents an atomic unit of state change. We distinguish three categories of components: edge components (addition or removal $\gamma$ of predicates $\rho$ between two entities $e_1$ and $e_2$), node components (addition or removal $\gamma$ of the predicate $\rho$ of entity $e$), and node transition components (transition from the previous predicate $\rho_{from}$ to new predicate $\rho_{to}$ of an entity $e$).

**Error Modeling from Signatures**   We categorize errors from two perspectives: structural and semantic. Structural errors concern the form of actions and include entity substitution (object replacement), predicate substitution (relation/attribute replacement), polarity inversion (add, remove or transition), omission, and hallucination. Semantic errors concern interpretation and are grouped into spatial relations (misplaced object positions), functional states (incorrect functionalities or status), material states (wrong physical properties), and agent interaction (misattributed agent actions). Both perspectives are based on comparing component sets of paired ground-truth and predicted signatures. For each pair, we compute set-level differences and classify components into missing (in ground truth only), matched (in both), and hallucinated (in prediction only). To support this categorization, we preprocess the signature dataset into structured data with these three groups of components, as outlined in Algorithm 2.

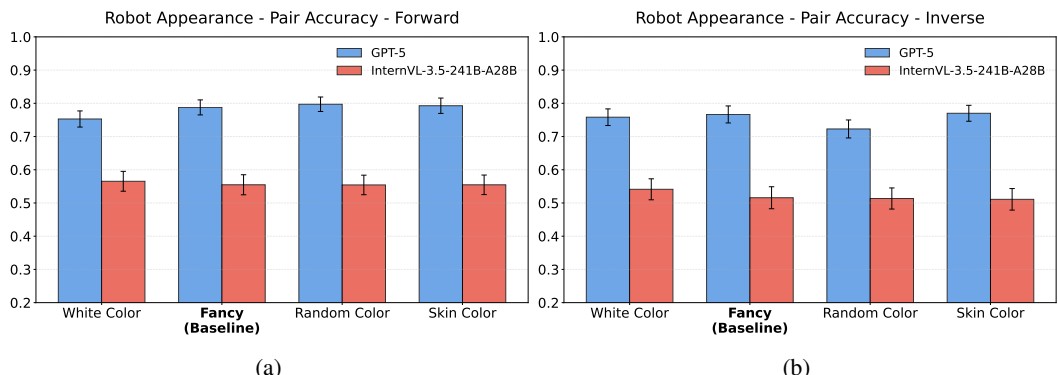

(a)                                                          (b)

Figure 34: **Probing robot appearance with GPT-5 and InternVL-3.5-241B-A28B.**(a) Forward dynamics; (b) Inverse dynamics. Bar plots report Pairwise Accuracy across four styles—White Color, Fancy (Baseline), Random Color, and Skin Color. Error bars denote ±SEM; the baseline tick is bolded.

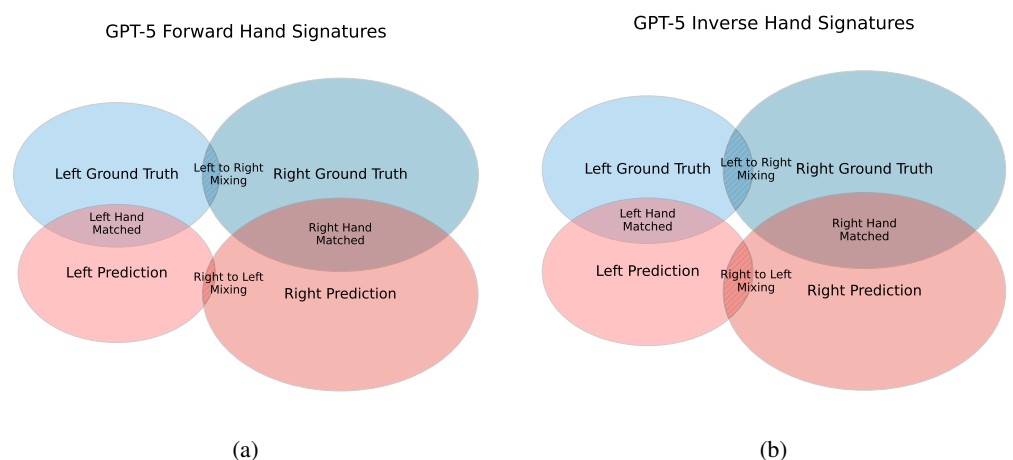

(a)                                                          (b)

Figure 35: Illustration of overlaps between ground truth and GPT-5 predictions sets for left and right hands related signatures in forward and inverse tasks. The size of ellipses project the total counts of signatures, and overlaps denote matched signatures (center regions) or mixing errors (cross-hand overlaps).

To categorize structural errors, we define criteria for each component type (edge, node, transition node). Entity Substitution occurs when entities differ while other fields match; Predicate Substitution when the predicate differs; and Polarity Inversion when only the operation (add/remove) differs. After pairwise classification, remaining unmatched ground-truth components are categorized to Omission, and unmatched predicted components as Hallucination.

After structural error categorization, each component is further labeled by semantic error type: Spatial Relations, Functional States, Material States, or Agent Interactions. Labeling uses a predefined mapping table that links all observed predicates to their semantic categories. When a component contains a listed predicate, the table is consulted to assign its semantic label. The overall workflow of error detection and categorization is illustrated in Algorithm 3.

**Handedness Asymmetry Error Modeling** To systematically capture handedness asymmetry, we compute for left- and right-hand components: precision (correct matches over predicted), recall (correct matches over ground truth), and the hand-mixing rate (the fraction of ground-truth left-hand components predicted as right, or vice versa).

For computing the hand-mixing rate, we define left–right mixing at the level of each signature-level difference (with missing, matched, and hallucinated components). If the missing set contains left- (or right-) hand usage, while the hallucinated set lacks the same hand but includes the opposite one, then

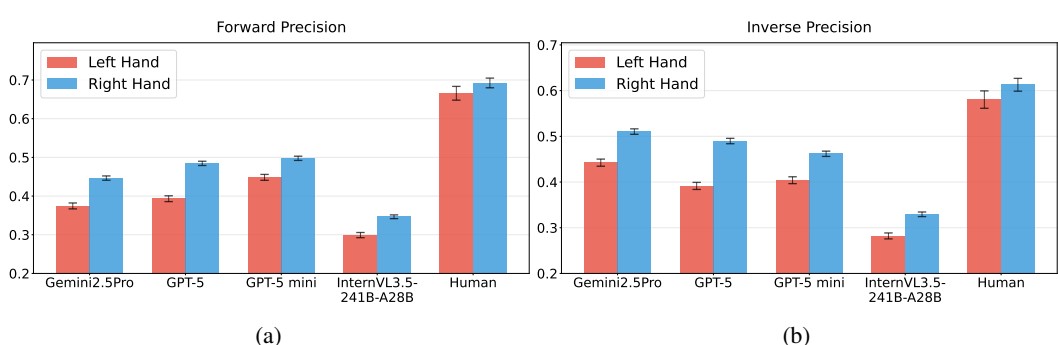

Figure 36: Precision of left/right hand related components prediction in (a) forward and (b) inverse tasks, with models Gemini2.5Pro, GPT-5, GPT-5 mini, InternVL3.5-241B-A28B, and Human. Error bars indicate the standard error (SE).

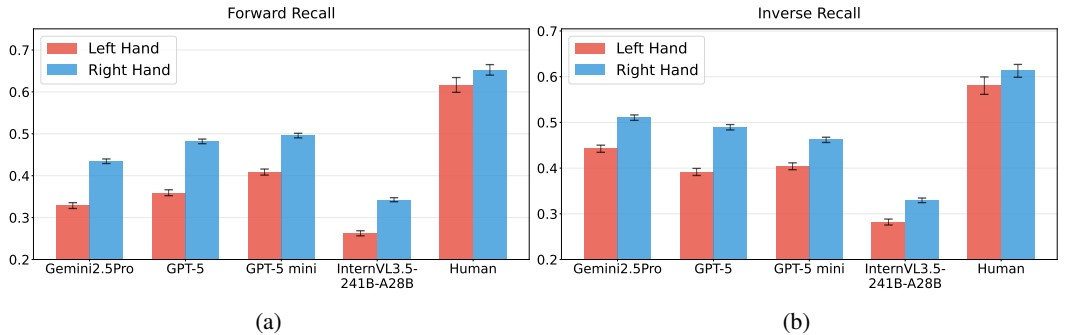

Figure 37: Recall of left/right hand related components prediction in forward task, with models Gemini2.5Pro, GPT-5, GPT-5 mini, InternVL3.5-241B-A28B and Human. Error bars indicate the standard error (SE).

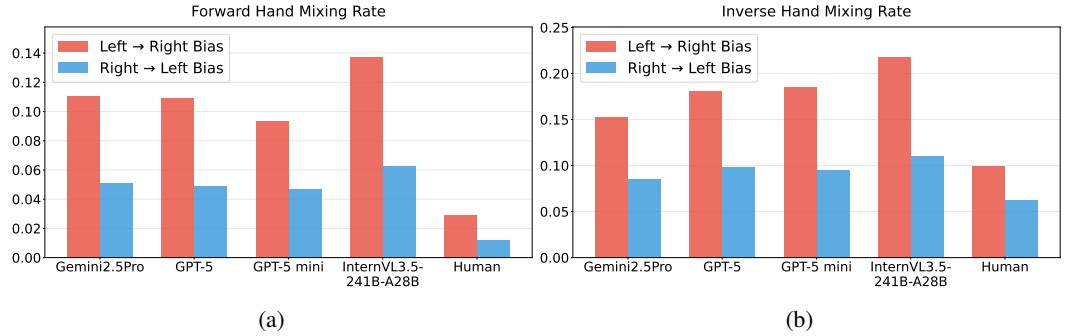

Figure 38: Hand-mixing rate, i.e.the ratio of left/right hand-mixing to all ground truth left/right and components in (a) forward and (b) inverse task, with models Gemini2.5Pro, GPT-5, GPT-5 mini, InternVL3.5-241B-A28B and Human. Error bars indicate the standard error (SE).

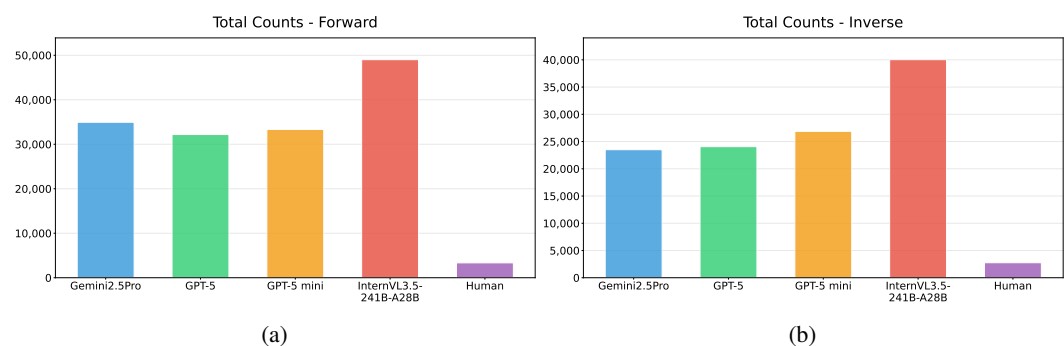

(a)                                                              (b)

Figure 39: The amount of total errors made by Gemini2.5Pro, GPT-5, GPT-5 mini, InternVL3.5-241B-A28B, and Human, under (a) forward tasks and (b) inverse tasks.

all missing components involving that hand are counted as left-to-right (or right-to-left) mixing, as outlined in Algorithm 4.

## C.2 STRUCTURAL ERROR ANALYSIS

We compared error patterns in forward and inverse tasks across Gemini-2.5 Pro, GPT-5, GPT-5 mini, InternVL-3.5-241B-A28B, and human predictions (Figures 39). All models showed markedly higher error rates than humans, with InternVL-3.5-241B-A28B producing the most errors overall.

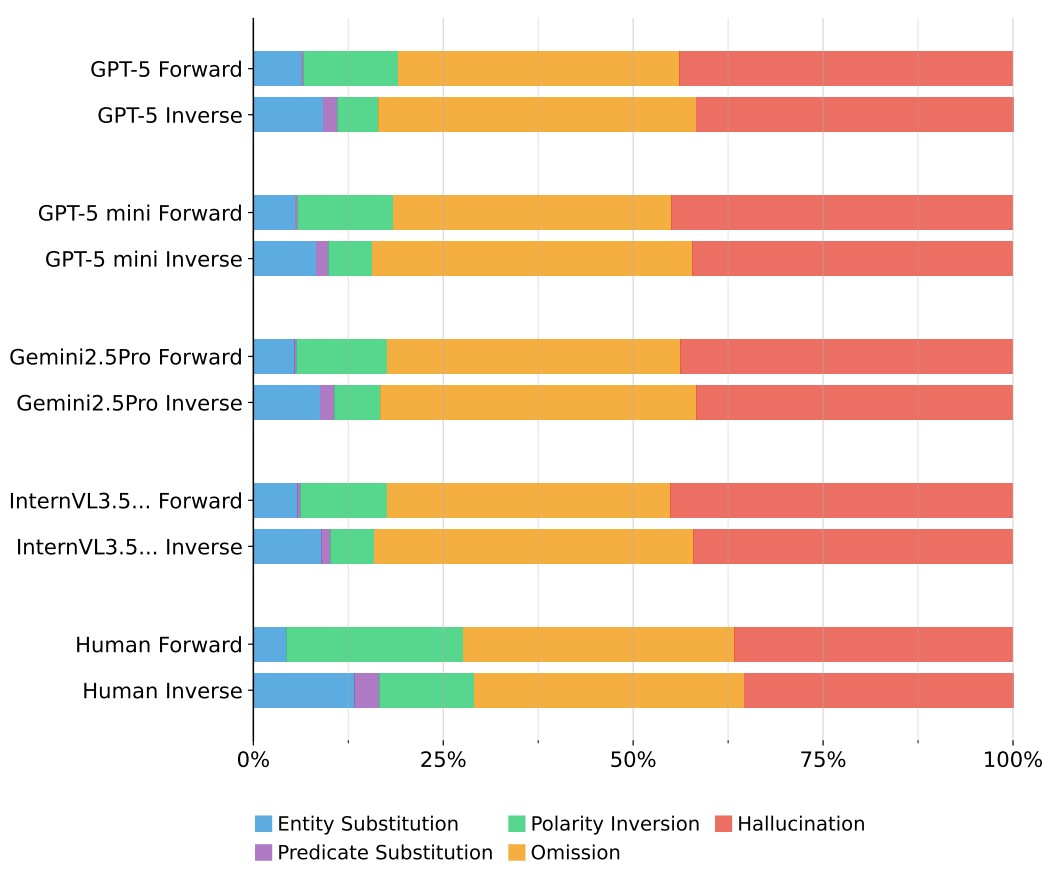

Figure 40: The structural error distributions of typical LLMs (GPT-5, GPT-5 mini, Gemini2.5Pro and InternVL3.5-241B-A28B (referred as InternVL3.5... in figure)) and Human-level prediction in both forward and inverse tasks.

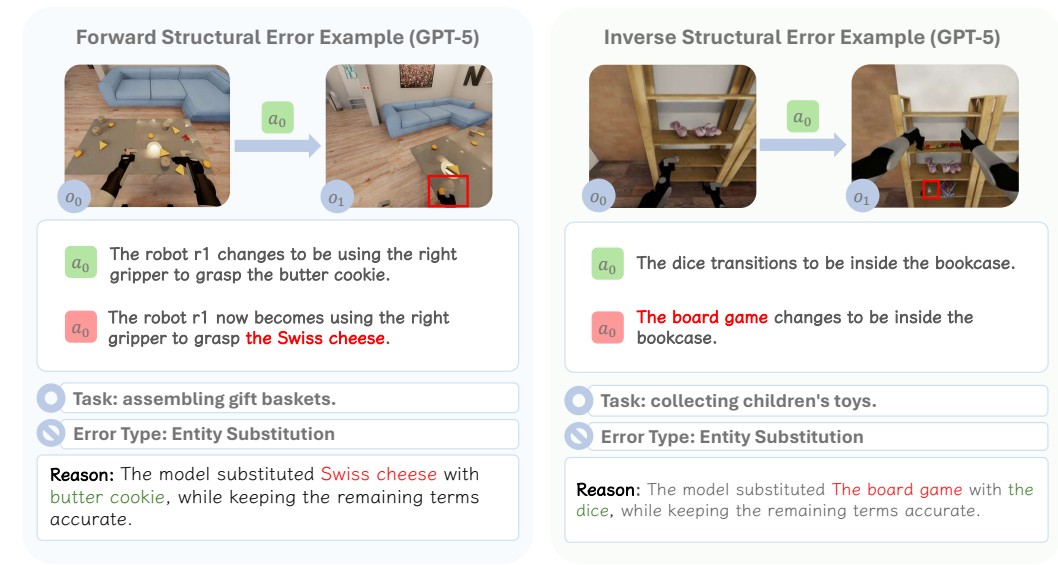

Figure 41: Example of structural error **Entity Substitution** by GPT-5 under forward and inverse tasks.

Using our component-level categorization, we analyzed structural error distributions across Gemini-2.5 Pro, GPT-5, GPT-5 mini, InternVL-3.5-241B-A28B, and Humans (Figure 40). The results are consistent: Omission and Hallucination each account for over 40% of errors, while Predicate Substitution is rare. This suggests a prevalence of coverage failures (missing or spurious components) rather than fine-grained reasoning. Notably, humans show relatively more Entity Substitution, Predicate Substitution, and Polarity Inversion, highlighting a potential difference in error patterns between humans and models.

Cross-condition comparison shows that Omission dominates in the forward setting, whereas in the inverse setting Omission and Hallucination are balanced, with Predicate Substitution also more frequent. For qualitative analysis, we sampled representative GPT-5 cases for each structural error type, stratified by forward and inverse tasks (Figure 41, 42, 43, 44, 45).

### C.3 SEMANTIC ERROR ANALYSIS

In our semantic error analysis (Figure 46), all systems—Gemini-2.5 Pro, GPT-5, GPT-5 mini, InternVL-3.5-241B-A28B, and humans—show a similar pattern: errors are concentrated in Spatial Relations and Agent Interaction, reflecting difficulties in reasoning about object positions and agent actions (e.g., left/right-hand grasping). A task-dependent asymmetry also appears: spatial-relations errors are more common in forward tasks, while agent-interaction errors are higher in inverse tasks. For illustration, we sample representative GPT-5 cases for each semantic category under both settings (Figures 47, 48, 49, 50).

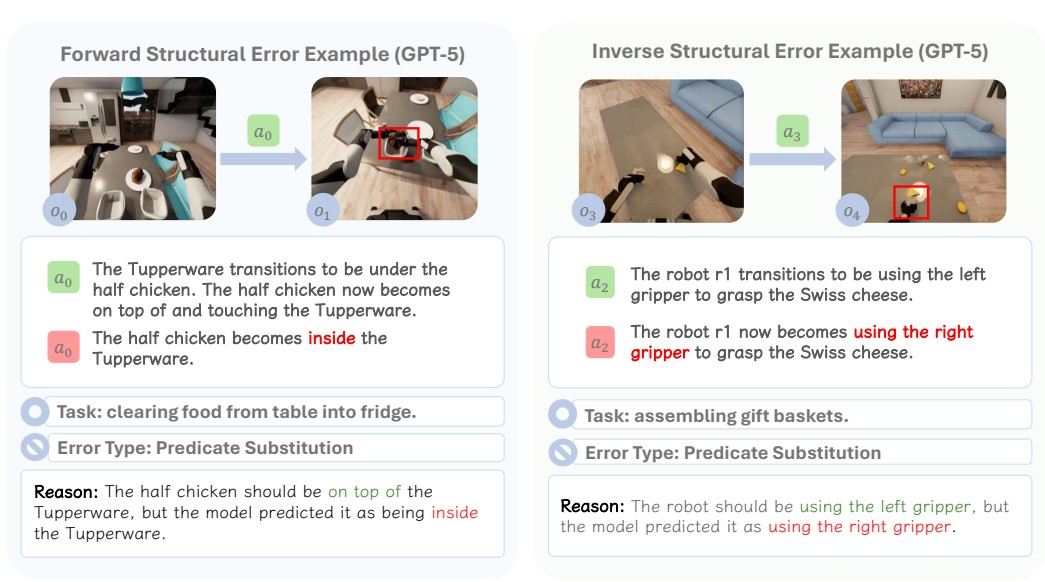

Figure 42: Example of structural error **Predicate Substitution** by GPT-5 under forward and inverse tasks.

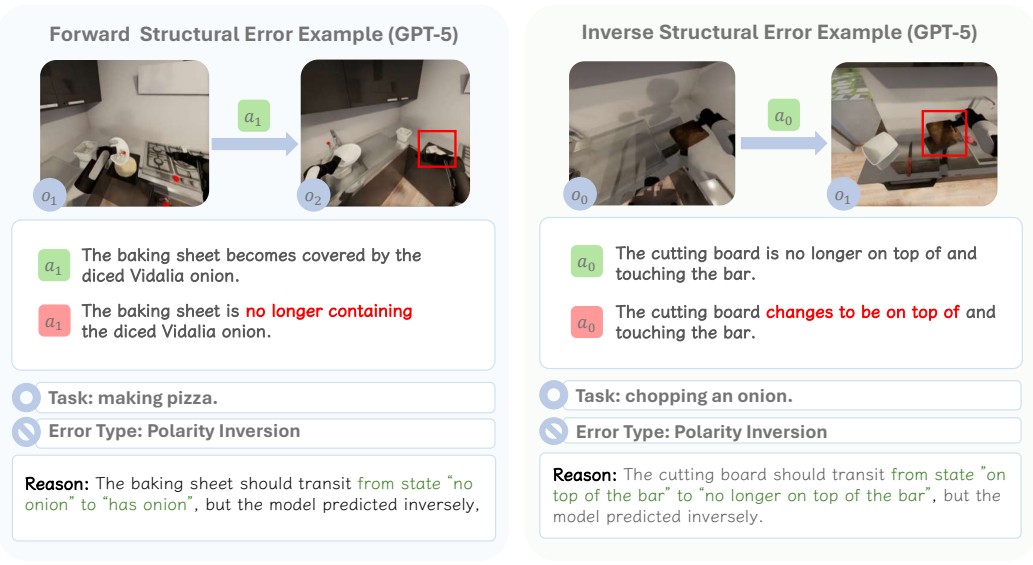

Figure 43: Example of structural error **Polarity Inversion** by GPT-5 under forward and inverse tasks.

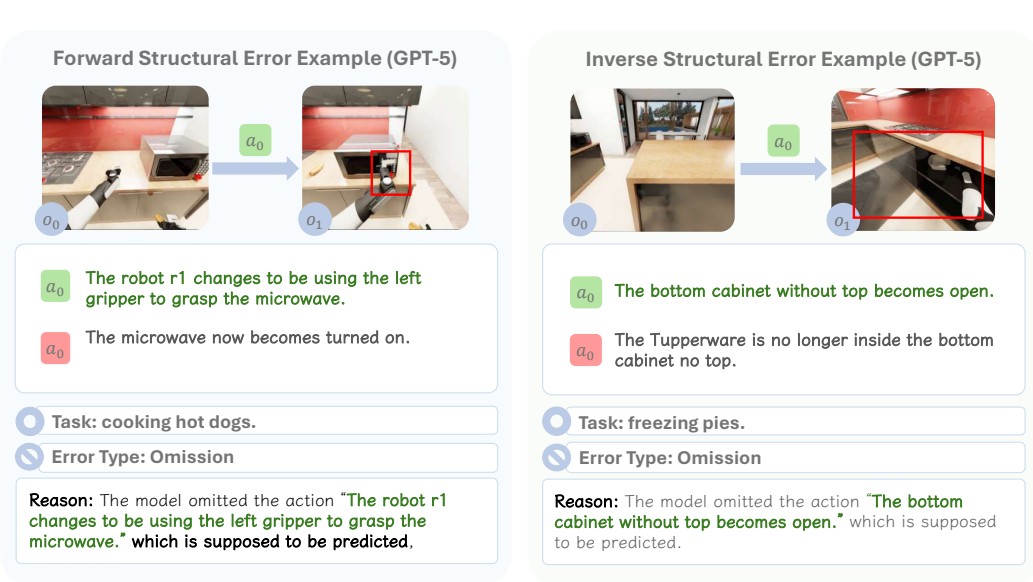

Figure 44: Example of structural error **Omission** by GPT-5 under forward and inverse tasks.

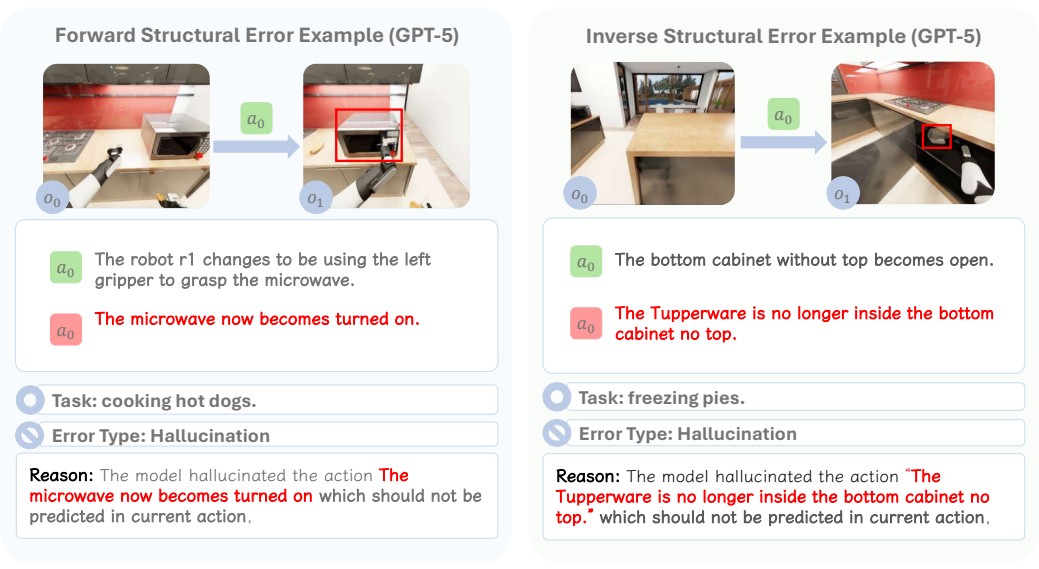

Figure 45: Example of structural error **Hallucination** by GPT-5 under forward and inverse tasks.

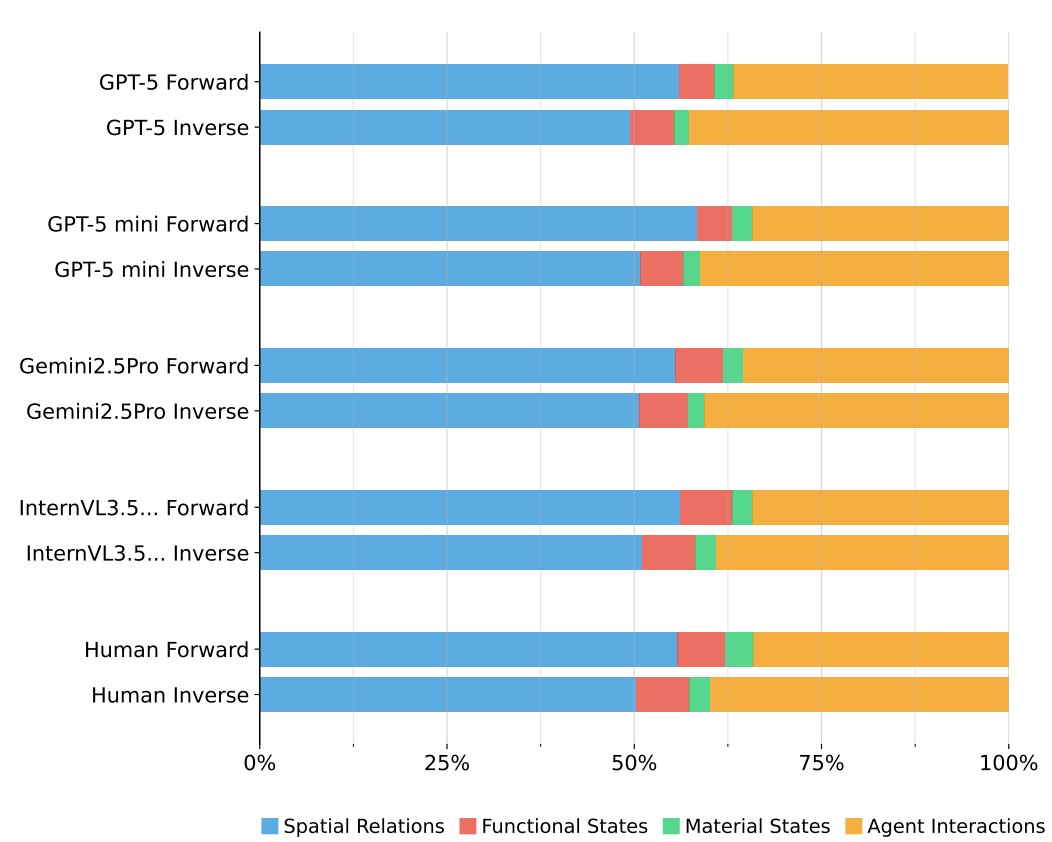

Figure 46: The semantic error distributions of typical LLMs (GPT-5, GPT-5 mini, Gemini2.5Pro and InternVL3.5-241B-A28B (referred as InternVL3.5... in figure)) and Human-level prediction in both forward and inverse tasks.

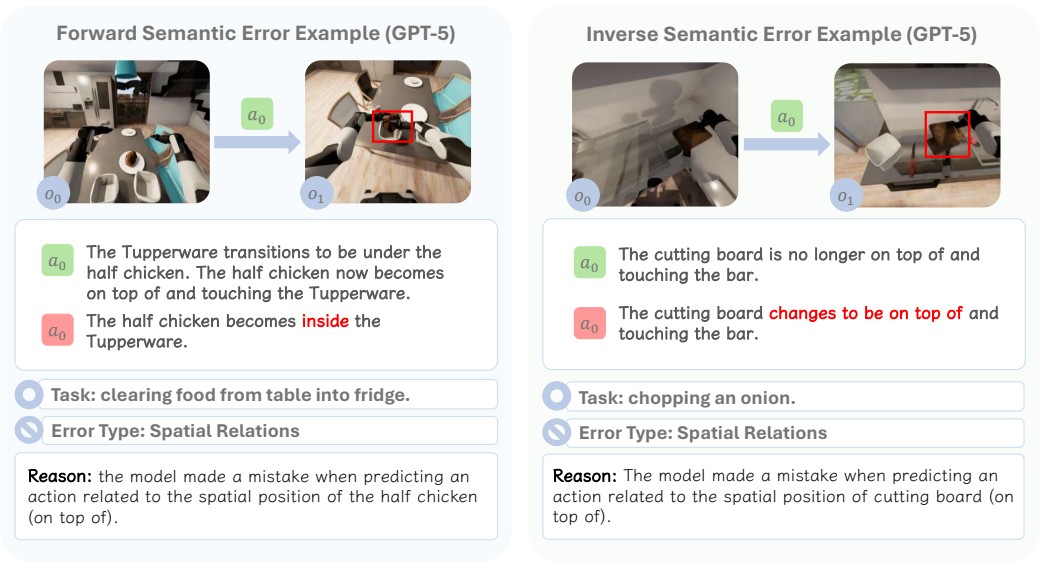

Figure 47: Example of semantic error **Spatial Relations** by GPT-5 under forward and inverse tasks.

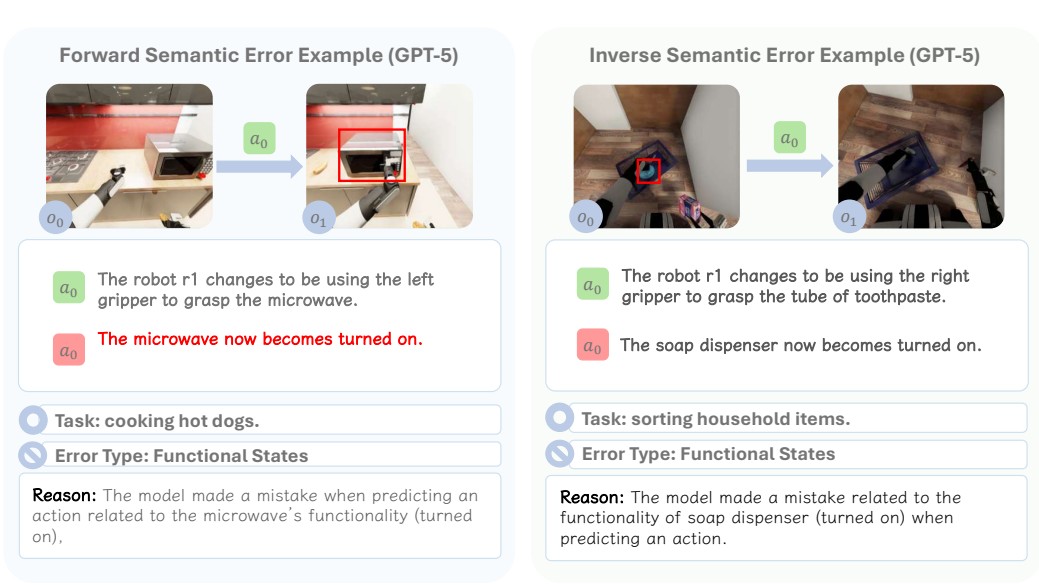

Figure 48: Example of semantic error **Functional States** by GPT-5 under forward and inverse tasks.

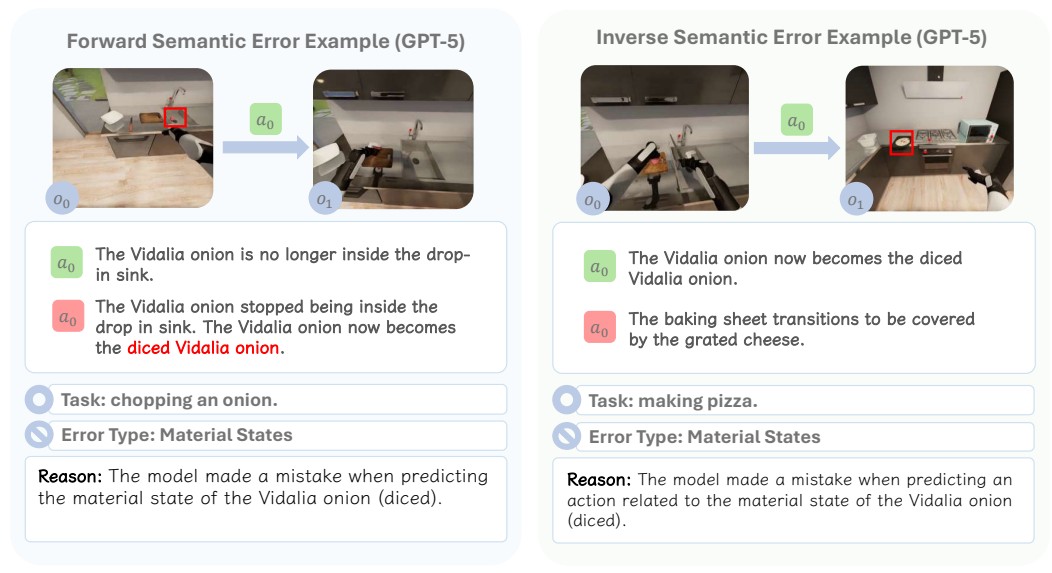

Figure 49: Example of semantic error **Material States** by GPT-5 under forward and inverse tasks.

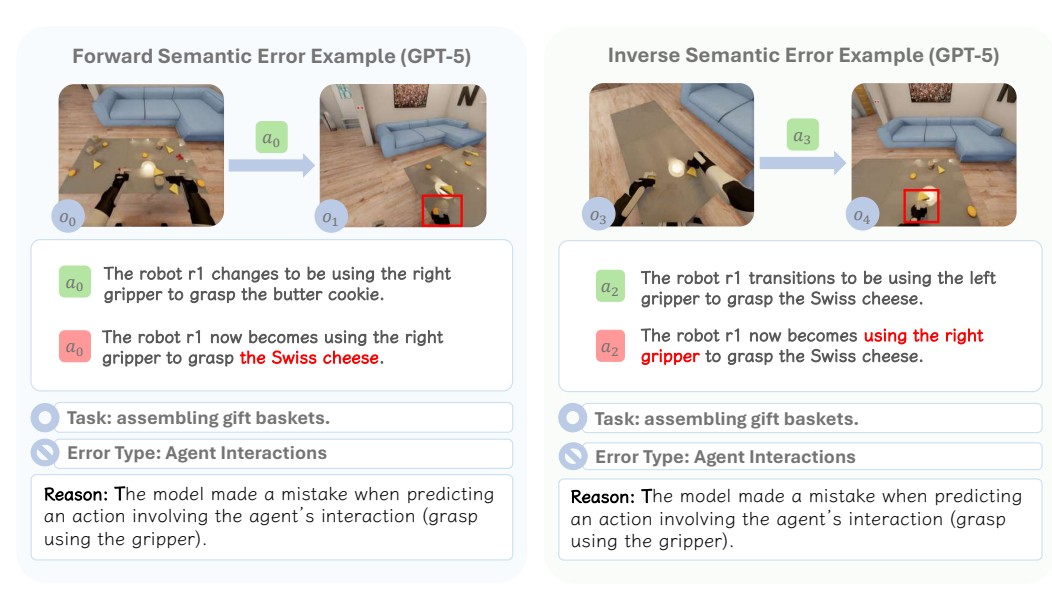

Figure 50: Example of semantic error **Agent Interactions** by GPT-5 under forward and inverse tasks.

---

**Algorithm 2:** Action-level Parsing of Signatures Data

---

**Input:** Dataset of signatures $\mathcal{D}_{sig}$, each with ground-truth signatures $a_{gt}^{sig}$ and predicted signatures $a_p^{sig}$

**Output:** Data of signatures $\mathcal{D}'_{sig}$ with missing, matched and hallucinated components

**Signatures filtering**: $\mathcal{D}'_{sig} \leftarrow \emptyset$

**foreach** $(a_{gt}^{sig}, a_p^{sig}) \in \mathcal{D}_{sig}$ **do**

    **if** $|a_{gt}^{sig}| = |a_p^{sig}|$ **then**

        add $(a_{gt}^{sig}, a_p^{sig})$ to $\mathcal{D}'_{sig}$

    **else**

        discard $(a_{gt}^{sig}, a_p^{sig})$

**Action-pairwise Comparison**: **foreach** $(a_{gt}^{sig}, a_p^{sig}) \in \mathcal{D}'_{sig}$ **do**

    /* $c_{mi}$: missing components, $c_{ma}$: matched components, $c_h$: hallucinated components     */

    $c_{mi} \leftarrow \emptyset, c_{ma} \leftarrow \emptyset, c_h \leftarrow \emptyset$

    **foreach** $(c_{gt}, c_p) \in (a_{gt}^{sig}, a_p^{sig})$ **do**

        **if** $c_{gt} = c_p$ **then**

            add $c_{gt}$ to $c_{ma}$

        **else**

            add $c_{gt}$ to $c_{mi}$

            add $c_p$ to $c_h$

    add $(c_{mi}, c_{ma}, c_h)$ to $\mathcal{D}'_{sig}$, discard $(a_{gt}^{sig}, a_p^{sig})$

**return** $\mathcal{D}'_{sig}$

---

---

**Algorithm 3:** Action-level Structural and Semantic Error Categorization

---

**Input:** Parsed signatures dataset $\mathcal{D}'_{sig}$, predicates $preds$
**Output:** Categorized errors dataset $\mathcal{D}_{err}$
**Structural errors categorization**: $PI \leftarrow \emptyset, \quad PS \leftarrow \emptyset, \quad ES \leftarrow \emptyset, \quad OM \leftarrow \emptyset, \quad HA \leftarrow \emptyset$
**foreach** $a^{sig} \in \mathcal{D}'_{sig}$ **do**
    $(C_{mi}, C_h) \leftarrow (c_{mi}(a^{sig}), \ c_h(a^{sig}))$
    $(c_{mi}, c_h) \leftarrow \texttt{FindPairwiseErrors}\,(C_{mi}, C_h, \text{polarity inversion})$;
    **if** $(c_{mi}, c_h) \neq \emptyset$ **then**
        add $(c_{mi}, c_h)$ to $PI$
        remove $(c_{mi}, c_h)$ from $a^{sig}$
    $(c_{mi}, c_h) \leftarrow \texttt{FindPairwiseErrors}\,(C_{mi}, C_h, \text{predicate substitution})$;
    **if** $(c_{mi}, c_h) \neq \emptyset$ **then**
        add $(c_{mi}, c_h)$ to $PS$
        remove $(c_{mi}, c_h)$ from $a^{sig}$
    $(c_{mi}, c_h) \leftarrow \texttt{FindPairwiseErrors}\,(C_{mi}, C_h, \text{entity substitution})$;
    **if** $(c_{mi}, c_h) \neq \emptyset$ **then**
        add $(c_{mi}, c_h)$ to $ES$
        remove $(c_{mi}, c_h)$ from $a^{sig}$
    **foreach** $c_{mi} \in C_{mi}$ **do**
        add $mi$ to $OM$
    **foreach** $c_h \in C_h$ **do**
        add $h$ to $HA$
/* PI: Polarity Inversion, PS: Predicate Substitution, ES:
   Entity Substitution, OM: Omission, HA: Hallucination    */
$\mathcal{D}_{err} \leftarrow (PI, PS, ES, OM, HA)$
**Semantic errors labeling**: **foreach** $c$ $in$ $\mathcal{D}_{err}$ **do**
    **foreach** $pred$ $in$ $preds$ **do**
        **if** $pred \in c$ **then**
            label $c$ with $\texttt{SemanticError}(pred)$

**return** $\mathcal{D}_{err}$

---

---

**Algorithm 4:** Dataset-Level Detection of Left–Right Hand Confusion

---

**Input:** Dataset of signature-level differences $\mathcal{D}_{diff} = \{(c_{mi}, c_{ma}, c_h)\}$
**Output:** Confusion dataset $\mathcal{D}_{hand} = \{(\mathcal{D}_{l2r}, \mathcal{D}_{r2l})\}$
**Left to right hand confusion**: $\mathcal{D}_{l2r} \leftarrow \emptyset$
**foreach** $(c_{mi}, c_{ma}, c_h) \in \mathcal{D}_{diff}$ **do**
   **if** $\exists m \in c_{mi}$ *that involves left hand* **then**
      **if** $\exists h \in c_h$ *that involves left hand* **then**
         **continue**,
      **else if** $\exists h \in c_h$ *that involves right hand* **then**
         **foreach** $m \in c_{mi}$ **do**
            **if** $m$ *involves left hand* **then**
               add $m$ to $\mathcal{D}_{l2r}$

**Right to left hand confusion**: $\mathcal{D}_{r2l} \leftarrow \emptyset$
**foreach** $(c_{mi}, c_{ma}, c_h) \in \mathcal{D}_{diff}$ **do**
   **if** $\exists m \in c_{mi}$ *that involves right hand* **then**
      **if** $\exists h \in c_h$ *that involves right hand* **then**
         **continue**,
      **else if** $\exists h \in c_h$ *that involves left hand* **then**
         **foreach** $m \in c_{mi}$ **do**
            **if** $m$ *involves right hand* **then**
               add $m$ to $\mathcal{D}_{r2l}$

$\mathcal{D}_{hand} \leftarrow (\mathcal{D}_{l2r}, \mathcal{D}_{r2l})$
**return** $\mathcal{D}_{hand}$

---

## D    Additional Related Work

**Embodied Cognition.**    The embodiment hypothesis holds that intelligence emerges from the ENACT coupling of perception and action within a physical, social, and linguistic world (Smith & Gasser, 2005). Meaning is not passively acquired but constructed through coherent bindings of spatial perception, physical interaction, and linguistic understanding—capacities that scaffold one another rather than develop in isolation. Philosophical and cognitive traditions echo this stance, emphasizing that cognition is shaped by morphology and situated experience (Clark, 1998; Varela et al., 2017; Gibson, 2014; Barsalou, 1999). This view motivates evaluating embodied agents not merely as robotic systems but as probes into the conditions under which general intelligence can arise.

**World Modeling.**    World modeling operationalizes embodied cognition by learning action-conditioned dynamics that support prediction and planning (Ha & Schmidhuber, 2018; Hafner et al., 2019). Beyond forward modeling, inverse reasoning about the actions that produced observed outcomes, under partial observability, connects naturally to a POMDP framing (Åström, 1965; Sutton et al., 1999). Recent methods explore transformer-based dynamics (Chen et al., 2022) and propose diagnostics for temporal abstraction and causality (e.g., CATER and CLEVRER) (Girdhar et al., 2020; Yi et al., 2019). Several benchmarks extend this direction. Dang et al. (2025) introduce ECBench under the theme of embodied cognition, but their evaluation focuses on multimodal perception QA without grounding in MDP theory or assessment of interactive dynamics, leaving the role of action and consequence largely unexplored. By contrast, Gao et al. (2025) examines whether vision–language models acquire internal models of the world, studying both spatial reasoning and physical interactions in contrived static setups. While non-egocentric, this work provides useful insight into how large models capture specific dimensions of reasoning. More recently, Chen et al. (2025) evaluates sequence-level coherence by having models predict the order of intermediate clips given only an initial and final state. While this assesses high-level planning, our benchmark, EN-ACT, differs in several fundamental ways to probe a more fine-grained understanding of interaction dynamics. First, their work lacks a clear action space, defining actions as video clips, which can lead to inconsistent semantic granularity. Second, their prediction of the entire intermediate sequence tests one-shot planning rather than a step-by-step, causal understanding of how actions lead to state changes. Furthermore, their evaluation is limited to forward prediction, whereas ENACT also probes inverse modeling. Finally, ENACT is built with a scalable data generation pipeline specifically designed to serve as a controlled proxy for probing the properties of VLM-based embodied agents. In doing so, it complements prior benchmarks by grounding embodied world modeling in long-horizon, fine-grained dynamics.

**Vision–Language Models for Embodied Agents.**    Scaled foundation models advance visual–linguistic reasoning, yet their primarily disembodied training raises a natural question of whether embodied cognition emerges without interaction (OpenAI, 2025; DeepMind, 2025; Anthropic, 2025). Robotics integrates VLMs as planners and policies—grounding language in affordances and control (Ahn et al., 2022; Huang et al., 2023b; 2022; Liang et al., 2022a; Driess et al., 2023; Zitkovich et al., 2023; Kim et al., 2024; Team et al., 2024). Parallel embodied agents in simulation assess navigation and interaction (Anderson et al., 2018; Das et al., 2018; Shridhar et al., 2020; Padmakumar et al., 2022; Mees et al., 2022; Fan et al., 2022), while egocentric corpora broaden sensorimotor coverage (Damen et al., 2018; Grauman et al., 2022). Complementary benchmarks probe isolated facets: spatial perception in static scenes (Ramakrishnan et al., 2024; Yin et al., 2025), contrived physical interactions (Yi et al., 2019; Bakhtin et al., 2019), or purely linguistic reasoning (Li et al., 2024b), whereas ENACT unifies these elements via egocentric trajectories and evaluates consequence-aware world modeling over extended horizons.

## E    The Use of Large Language Models

We used large language models (LLMs), including Google's Gemini 2.5 Pro and OpenAI's GPT-5, as auxiliary tools to assist with writing, editing, and conducting the literature review for this manuscript. All content was critically revised and fact-checked by the human authors to ensure its scientific validity and originality. The authors are fully responsible for all statements and conclusions presented in this paper. Specifically, we use LLMs for polishing our wording and writing, and we use LLMs to retrieve several related works.

