# OpenReview forum: "ENACT: Evaluating Embodied Cognition with World Modeling of Egocentric Interaction"
_ICLR.cc/2026/Workshop/AFAA — Submitted to AFAA 2026_

### Official Review · Reviewer_z9vs · 2026-02-18
**Review: ENACT — Evaluating Embodied Cognition with World Modeling of Egocentric Interaction**

**Rating:** 5
**Confidence:** 4

**Summary:**

ENACT is a benchmark for probing embodied cognition in VLMs through egocentric world modeling. Grounded in a POMDP framework, it poses two sequence reordering tasks: forward world modeling (ordering future state observations given actions) and inverse world modeling (ordering actions given state observations). The benchmark contains 8,972 QA pairs from BEHAVIOR simulator trajectories with symbolic scene graph representations, evaluating 29 VLMs across step lengths 3–10. Key findings: (1) VLMs lag significantly behind humans, with performance degrading as horizon increases; (2) inverse modeling is consistently easier than forward; (3) models exhibit right-hand bias and sensitivity to non-human camera configurations.

**Strengths:**

Strengths
1. Well-motivated and theoretically grounded. The POMDP framing is principled, the connection to embodied cognition is clearly articulated, and the choice of forward/inverse dynamics as complementary probes of world understanding is conceptually sound.

2. Scalable data generation pipeline. The combinatorial sampling from segmented frames (Algorithm 1) is clever and well-documented. The use of symbolic scene graphs as abstract actions is a pragmatic design choice that enables clean experimental control while maintaining grounding in real robotic predicates.

3. Comprehensive evaluation and ablations. 29 VLMs tested, human baselines with strong IAA (α=0.83), controlled probing experiments on image realism, camera FOV/height, and robot appearance. The error analysis (structural + semantic) is thorough and provides interpretable failure modes.

4. Honest empirical reporting. The paper is transparent about what the benchmark measures and does not claim more than the data supports. The acknowledgment of limitations (simulator-only, computational cost) is appropriate.

5. Reproducible. Code, data generation pipeline, prompts, and evaluation scripts are documented in the appendix. The online verifier allowing multiple valid orderings is a principled design choice.

**Weaknesses:**

Weaknesses

1. Forward > Inverse asymmetry is mechanistically underexplained.
The paper reports that inverse modeling is consistently easier (Section 3.1, Table 1), attributes this to inverse being a "constrained matching problem" vs. forward being an "unconstrained simulation problem," but provides no ablation or analysis to validate this explanation.

Alternative hypotheses not tested:

* Is the asymmetry due to VLMs being better at vision→text alignment (inverse) than text→vision prediction (forward)?

* Does the scene graph action representation advantage inverse tasks because textual actions are easier to order than visual states?

* Is forward harder because multiple valid visual futures exist for a given action sequence, but actions are more deterministic given observations?

The paper needs either: (a) an ablation varying the modality of states/actions (e.g., text-described states instead of images for forward), or (b) a control task where both forward and inverse use the same modalities, to isolate whether the difficulty gap is inherent to the reasoning direction or an artifact of the task design.

2. Human performance ceiling is unexplained and may indicate task artifacts. Humans achieve 93–97% pairwise accuracy (Table 1) even at L=10, showing no degradation with horizon length. This is surprising given that:

The tasks require tracking 10-step causal chains. Symbolic actions like Remove OnTop(pen, desk) are unnatural descriptions. Images are 512×512 simulator renderings with complex scenes

Two possibilities:

* The tasks are actually easy for humans because scene graph changes are visually salient, suggesting VLM failures reflect vision or attention deficits rather than world modeling per se.
* Humans are leveraging task-specific heuristics (e.g., object permanence, spatial continuity) that the symbolic framing makes explicit.

The paper should either: (a) analyze why humans don't degrade with L, or (b) discuss what this implies about the benchmark's difficulty and what it measures.


3. Right-hand bias analysis is descriptive, not explanatory. Section 3.4 reports higher precision/recall for right-hand vs. left-hand predicates and attributes this to "human handedness asymmetry" in training data. However:

* No analysis of whether BEHAVIOR demonstrations themselves are right-biased
* No ablation testing if left/right labels are swapped (would models still favor "right" if it were relabeled?)
* No investigation of whether the bias emerges from vision (e.g., right hand more visible) vs. language priors

4. Limited scope: simulator-only, single-task-type, no failure recovery.

* All data is from BEHAVIOR simulator — no real-world validation
* Tasks are always solvable by reordering — no test of detecting invalid or impossible sequences
* No evaluation of online replanning or error recovery when world models fail

---

### Official Review · Reviewer_si9H · 2026-02-21
**Benchmark for evaluating Vision-Language Models with a note on Emboddied Biases**

**Rating:** 2
**Confidence:** 3

**Summary:**

This paper introduces ENACT, a benchmark for evaluating Embodied Cognition (EC) in Vision Language Models (VLMs).


The authors motivate their benchmark with the observation that EC is usually learned from continuous sensorimotor interaction with the world, which is missing in the data used to train VLMs. Therefore, they propose a POMDP-based benchmark to test VLMs on two tasks: forward world modeling (predicting future states from actions) and inverse world modeling (inferring actions from states). The tasks boil down to ordering given observations or actions. Key-frame trajectories as well as 8972 QA pairs are generated from the BEHAVIOR data simulator (Li et al., 2024). Two evaluation metrics for the resulting permutations are used: task and pairwise accuracy.


The authors test 29 VLMs in total, comparing them for prediction (ordering) tasks with lengths between 3 and 10. Key results are: VLMs fall short of humans on the proposed metrics, VLMs performance drops rapidly with the length of the prediction as opposed to humans, inverse modeling performance is higher than forward modeling performance for VLMs.


Two possible embodied biases are tested: robots’ self-awareness regarding their appearance and handedness asymmetry in robots’ interactions with the world. VLMs are robust wrt their appearance (color), but do exhibit preference for right-handedness in their actions.
The paper concludes with an error analysis, discussion of related work and limitations.

**Strengths:**

The paper tackles two fairness-related questions: are there embodied biases concerning appearance self-awareness and handedness in the considered VLCs. The answer to the former is conclusively negative (changing robots’ color does not change world modeling performance), to the latter --- positive (robots’ actions show preference for right handedness). Such two observations made in a large scale study are novel.


The paper is well organized and easy to follow. It provides a thorough statistical analysis for the proposed metrics. The selection of state of the art VLMs is exhaustive. The key findings are relevant to the posed question (is there a gap between EC and the way VLMs are trained presently). The proposed benchmark can be reused by adding other models to it.

**Weaknesses:**

The short note on robot appearance and handedness (Section 3.4) is not followed by any remarks on how to deal with the discovered handedness bias (or any potential biases not tested at this point). No further discussion on how to use the ENACT benchmark or the BEHAVIOR data simulator for testing fairness ensuring algorithms is provided. The remainder of the paper (outside of Section 3.4) appears to be largely outside of the scope of this workshop.


As a general remark, I was wondering if reducing “prediction” and “inferring” of states and actions to sequence reordering does not relax the measure of embodied cognition too much. In other words, when thinking about predicting future states or inferring past actions, one would expect (for both humans and VLMs) something more than ordering given objects, perhaps a task with a generative component.

---

### Official Review · Reviewer_d8hf · 2026-02-21
**ENACT Review**

**Rating:** 3
**Confidence:** 4

**Summary:**

This paper introduces ENACT, a benchmark for evaluating embodied cognition in VLMs through world modeling from egocentric interaction. Using a POMDP framework, the paper introduces two sequence reordering tasks: (i) forward world modeling and (ii) inverse world modeling. The dataset contains 8,972 QA pairs from 29 household activities in the BEHAVIOR simulator, with trajectory lengths of 3 to 10 steps. The paper evaluates 29 VLMs and human annotators, finding a significant and widening performance gap as horizons increase. Probing experiments investigate sensitivity to image realism, camera configurations, and embodied biases including handedness asymmetry.

**Strengths:**

- The sequence reordering framing elegantly mitigates the problem of designing unbiased distractors for long-horizon interactions and scales naturally with trajectory length.

- The forward/inverse framing provides complementary views on world modeling.

- The controlled studies on camera FOV, camera height, and handedness asymmetry are the paper's most scientifically interesting contributions, revealing concrete properties of how VLMs process embodied visual information beyond standard benchmark scores.

- The evaluation is torough. 29 VLMs, human baselines with proper inter-annotator agreement ($\alpha = 0.83$), and a well-designed error analysis that decomposes failures into interpretable structural and semantic categories.

- The DAG-based trajectory sampling algorithm is technically sound and well-documented.

**Weaknesses:**

- The paper motivates itself through embodied cognition theory but the actual tasks are standard forward/inverse dynamics prediction. The gap between the philosophical framework and what is operationally tested (matching symbolic state-change descriptions to visual differences in simulator screenshots) could use more careful discussion.

- The benchmark draws from only 29 activities with one trajectory each. Combinatorial sampling generates many QA pairs but does not add new visual or semantic diversity. It would be useful to see ablations varying the number of source trajectories.

- The tasks are fundamentally temporal sequence ordering with multimodal inputs. Comparing ENACT to relevant work on event ordering, script knowledge, and procedural reasoning in LLMs would help clarify what is specifically "embodied" about the benchmark versus general temporal understanding.

- Using GPT-image-1 style transfer as a "realistic" proxy is creative but primarily tests robustness to texture/lighting changes rather than genuine sim-to-real transfer.

- Some of the main findings (degradation with longer horizons, inverse easier than forward, models lagging humans) are largely expected. The probing experiments produce more novel insights but are positioned as secondary rather than central.

---

### Meta-Review · Area_Chair_HYEv · 2026-02-21

**Recommendation:** Reject
**Confidence:** 5

**Metareview:**

The paper does not respect the dual submission policy of the workshop as it has already been accepted to ICLR 2026 as a poster, as pointed by Reviewer si9H. Thus te paper therefore cannot be included in the workshop.

---

### Decision · Program_Chairs · 2026-03-02

Reject